# The Easy, the Hard, and the Learnable:
# Confidence and Difficulty-Adaptive Policy Optimization for LLM Reasoning

Zhanke Zhou [1 2 *]  Xiangyu Lu [1 *]  Chentao Cao [1]  Brando Miranda [2]  Tongliang Liu [3]  Bo Han [1]  Sanmi Koyejo [2]

## Abstract

RL with verifiable rewards can substantially improve LLM reasoning, yet standard GRPO-style training often treats *easy*, *hard*, and *learnable* questions alike through uniform sampling and weighting, leading to inefficient compute allocation. We study GRPO by tracking token log-probabilities, group-normalized advantages, and the induced token-level update weights. This reveals three recurring dynamics as training proceeds: (1) *confidence inflation*, (2) *advantage contraction*, and (3) *hierarchical convergence*. These findings suggest that the utility of each update depends strongly on both question difficulty and the model's current competence. Motivated by this, we propose Confidence and Difficulty-adaptive Policy Optimization (CoDaPO), which assigns each question a bounded value from rollout confidence and empirical difficulty. CoDaPO then uses this value to reweight policy updates and resample high-value *learnable* questions within mini-batches, thereby increasing discovery within the learnable band under a fixed compute budget. Across twelve benchmarks, CoDaPO consistently improves accuracy over existing RL methods. Our code is publicly available at https://github.com/tmlr-group/CoDaPO.

## 1. Introduction

Reinforcement learning (RL) is increasingly used to improve large language models (LLMs) on *verifiable* reasoning tasks, such as mathematics and code generation, where model-generated trajectories can be automatically evaluated. PPO (Schulman et al., 2017) is a standard RL approach, but its learned value function adds overhead, motivating critic-free alternatives such as GRPO (Shao et al., 2024), which

*Equal contribution [1]TMLR Group, Department of Computer Science, Hong Kong Baptist University [2]Stanford University [3]Sydney AI Centre, The University of Sydney. Correspondence to: Bo Han <bhanml@comp.hkbu.edu.hk>.

*Proceedings of the 43rd International Conference on Machine Learning*, Seoul, South Korea. PMLR 306, 2026. Copyright 2026 by the author(s).

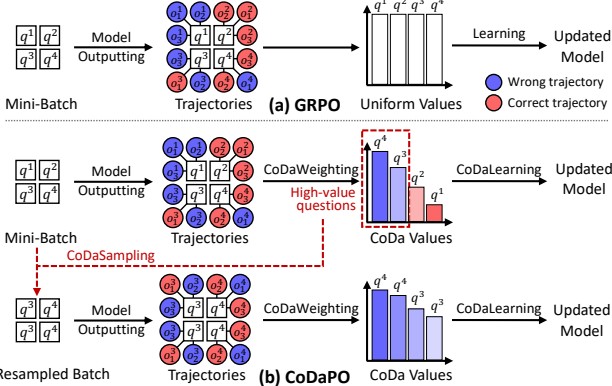

*Figure 1.* **Illustration of GRPO and CoDaPO. (a)** GRPO updates the model with uniformly weighted trajectories generated from a random mini-batch. **(b)** CoDaPO computes per-question *CoDa values* from confidence and difficulty, uses them to *weight* updates, and *resamples* high-value questions for a rollout-and-update step.

standardizes rewards within sampled trajectory groups and retains a PPO-style clipped objective.

Despite GRPO's strong empirical performance, its learning dynamics in sparse-reward, long-horizon reasoning remain under-investigated. GRPO relies on group-normalized advantages computed from a small number of sampled trajectories and updates the policy through a clipped surrogate objective; for difficult questions, progress may *hinge* on the rare event of sampling even a single correct trajectory. This raises a central question: *how do group normalization, clipping, and finite-sample discovery jointly determine which questions improve quickly, which ones saturate, and which ones remain bottlenecked as training proceeds?*

To answer this question, we analyze GRPO by monitoring training-time statistics, including token probabilities, group-normalized advantages, and the resulting token-level update weights. This analysis reveals three consistent patterns:

1. **Confidence inflation:** Model confidence concentrates near $100\%$ for both correct and incorrect trajectories, indicating degraded accuracy-confidence calibration.

2. **Advantage contraction:** As groups become more accurate, positive advantages shrink toward zero, while rare failures receive increasingly large negative advantages.

3. **Hierarchical convergence:** Easy questions quickly sat-

urate and yield vanishing gradients, whereas hard questions remain discovery-limited and improve slowly.

We attribute these patterns to two structural features of GRPO: (i) asymmetric clipping, which preserves upward probability drift while truncating sufficiently negative updates, and (ii) group normalization with binary rewards, which weakens positive learning signals as accuracy approaches one. Consequently, the utility of a model update is highly non-uniform and depends strongly on both question difficulty and the model's current competence.

These findings highlight a concrete inefficiency in standard GRPO-style training: *uniform sampling and near-uniform weighting can misallocate computation across easy, hard, and learnable questions.* Once a question is effectively solved, additional updates largely sharpen the model's distribution—often inflating confidence—with little improvement in correctness. In contrast, genuinely hard questions are often *discovery-limited*: with a small rollout group, the probability of observing even a single correct trajectory can be vanishingly small, so positive reinforcement rarely occurs. Under a fixed compute budget, improving hard-case performance therefore requires not only better per-sample objectives, but also a better computation-allocation strategy: deciding *which questions* deserve more trials and *which updates* should receive more emphasis.

Motivated by this diagnosis, we propose **Co**nfidence and **D**ifficulty-**a**daptive **P**olicy **O**ptimization (**CoDaPO**), a simple, data-centric method that integrates seamlessly with RL objectives by reweighting updates and directing sampling toward more informative questions (Fig. 1). Specifically, CoDaPO first applies **CoDaWeighting** to *assign* each question a bounded value using two signals readily available from sampled trajectories: *confidence* (mean token likelihood) and *difficulty* (group error rate). This value is then used in two complementary ways: **CoDaLearning** *rescales* policy-gradient updates via a value-weighted objective, concentrating gradient mass on questions with higher learning potential, while **CoDaSampling** *resamples* the top-$K$ questions in each mini-batch by value, repeating high-value questions to allocate more trials and increase the chance of discovering correct trajectories when successes are rare.

Conceptually, CoDaPO concentrates more computation on a "learnable band": it down-weights already-solved questions that provide little additional signal and avoids over-investing in extremely hard questions where learning is dominated by the absence of successful trajectories. Instead, it prioritizes *learnable* questions that are *challenging* enough to drive progress yet sufficiently *tractable* to yield reliable reinforcement. CoDaPO further improves efficiency by using token-level micro-averaging to eliminate implicit length penalties and removing KL-to-reference regularization to reduce overhead and encourage exploration, while preserving stability through bounded CoDa weights.

Experimentally, CoDaPO is proven effective on twelve reasoning benchmarks, consistently improving reasoning accuracy and generalization over existing RL methods. For example, compared to the base model Qwen2.5-Math-1.5B, CoDaPO increases accuracy from $30.63\%$ to $71.54\%$ on the in-domain MATH500 benchmark and from $18.78\%$ to $36.16\%$ on the out-of-domain OlympiadBench.

The main contributions of this work are:
- We provide empirical evidence and mathematical analysis of GRPO's training dynamics, explaining *confidence inflation*, *advantage contraction*, and *hierarchical convergence* in verifiable reasoning post-training (Sec. 3).
- We propose CoDaPO, a data-centric RL framework that reweights GRPO-style updates using per-question *confidence* and *difficulty*, thereby prioritizing informative trajectories and improving compute allocation (Sec. 4).
- We evaluate CoDaPO on twelve widely used reasoning benchmarks, showing that it consistently improves accuracy and generalization over GRPO and other RL baselines under comparable training budgets (Sec. 5).

## 2. Preliminaries

**Notation.** The training set $\mathcal{D}$ consists of question-answer pairs $(q, a)$. Given $q$, the policy $f_{\boldsymbol{\theta}}$ samples a trajectory $o \sim f_{\boldsymbol{\theta}}(\cdot \mid q)$ containing intermediate reasoning and a final answer. As in GRPO (Shao et al., 2024), we maintain three policies: the *current* policy $f_{\boldsymbol{\theta}}$ (updated every step), a frozen *behavior* policy $f_{\text{old}}$ (recent snapshot for sampling), and a *reference* policy $f_{\text{ref}}$ (older snapshot for divergence control).

**Group Relative Policy Optimization (GRPO).** Shao et al. (2024) simplifies PPO (Schulman et al., 2017) and estimates advantages by standardizing rewards within a sampled group. For each $q$, sample $G$ rollouts $\{o_i\}_{i=1}^{G} \sim f_{\text{old}}(\cdot \mid q)$ and compute rewards $\{r_i\}_{i=1}^{G}$, yielding advantage $\hat{A}_i \triangleq r_i - \text{mean}(\{r_j\}_{j=1}^{G})/\text{std}(\{r_j\}_{j=1}^{G})$. With the token-level importance ratio $\rho_{i,t} = f_{\boldsymbol{\theta}}(o_{i,t}|q,o_{i,<t})/f_{\text{old}}(o_{i,t}|q,o_{i,<t})$, GRPO maximizes a clipped objective with a KL penalty:

$$\mathcal{J}_{\text{GRPO}}(f_{\boldsymbol{\theta}}) \triangleq \mathbb{E}_{(q,a)\sim\mathcal{D}, \{o_i\}_{i=1}^{G}\sim f_{\text{old}}(\cdot|q)}\left[\frac{1}{G}\sum_{i=1}^{G}\frac{1}{|o_i|}\sum_{t=1}^{|o_i|}\right.$$

$$\left.\min\left(\rho_{i,t}\hat{A}_i, \text{clip}\left(\rho_{i,t}, 1-\epsilon, 1+\epsilon\right)\hat{A}_i\right) - \beta\mathbb{D}_{\text{KL}}[f_{\boldsymbol{\theta}}\|f_{\text{ref}}]\right].$$

**Related work.** Recent studies have further improved GRPO and related objectives, including DAPO (Yu et al., 2025), Dr. GRPO (Liu et al., 2025a), REINFORCE++ (Hu et al., 2025), CPPO (Lin et al., 2025b), GPG (Chu et al., 2026). Beyond text-only settings, several works extend these methods to multimodal reasoning (Zhou et al., 2025a; Huang et al., 2026) and logical reasoning (Xie et al., 2025b). A detailed discussion of these works is in Appendix A.

# 3. Training Dynamics of GRPO

In this section, we characterize GRPO training dynamics by (i) deriving the evolution of key sample statistics (Sec. 3.1) and (ii) validating them empirically (Sec. 3.2). We then analyze the mechanisms underlying these behaviors (Sec. 3.3).

These findings suggest that uniform weighting and sampling may squander compute on already-saturated easy questions, while truly hard questions are bottlenecked by the rarity of correct rollouts (i.e., exploration and discovery), motivating CoDaPO to reallocate updates and sampling accordingly.

## 3.1. Statistics of Training Samples

For each $(q, a) \in \mathcal{D}$, we sample a group of $G$ trajectories $\{o_i\}_{i=1}^{G} \sim f_{\text{old}}(\cdot \mid q)$, where $o_i = (o_{i,1}, \ldots, o_{i,|o_i|})$, and compute the following statistics on these samples.

**Probability (token-level).** Define $\ell_{\boldsymbol{\theta}}(o_{i,t}) \triangleq \log f_{\boldsymbol{\theta}}(o_{i,t} \mid q, o_{i,<t})$ and $\ell_{\text{old}}(o_{i,t}) \triangleq \log f_{\text{old}}(o_{i,t} \mid q, o_{i,<t})$, then the importance ratio $\rho_{i,t}$ is computed as

$$\rho_{i,t} \triangleq \frac{f_{\boldsymbol{\theta}}(o_{i,t} \mid q, o_{i,<t})}{f_{\text{old}}(o_{i,t} \mid q, o_{i,<t})} = \exp\big(\ell_{\boldsymbol{\theta}}(o_{i,t}) - \ell_{\text{old}}(o_{i,t})\big).$$

When needed, $p_{\boldsymbol{\theta}}(o_{i,t}) = \exp(\ell_{\boldsymbol{\theta}}(o_{i,t}))$. Note that all probabilities are post-softmax (log-probabilities, not logits). We work in log-probability space for numerical stability.

**Reward (trajectory-level).** We use a binary accuracy reward based on the final answer in $o_i$ and ground truth $a$:

$$r_i \triangleq \text{CorrectAnswer}(o_i, a) \in \{0, 1\},$$

where $r_i = 1$ if the predicted answer matches $a$.

**Advantage (trajectory-level).** GRPO estimates advantages by group-normalizing rewards:

$$\hat{A}_i \triangleq \frac{r_i - \text{mean}(\{r_j\}_{j=1}^{G})}{\text{std}(\{r_j\}_{j=1}^{G}) + \delta},$$

with a small $\delta > 0$ for numerical stability. We treat $\hat{A}_i$ as a constant w.r.t. $\boldsymbol{\theta}$ (stop-gradient), i.e., no backpropagation through the group statistics.

**Gradient (token-level).** GRPO maximizes a clipped surrogate objective with a per-token KL penalty:

$$\mathcal{J}_{\text{GRPO}}(f_{\boldsymbol{\theta}}) = \frac{1}{G} \sum_{i=1}^{G} \frac{1}{|o_i|} \sum_{t=1}^{|o_i|}$$

$$\underbrace{\min\big(\rho_{i,t}\hat{A}_i, \text{clip}(\rho_{i,t}, 1-\epsilon, 1+\epsilon)\hat{A}_i\big)}_{\text{clipped surrogate}} - \underbrace{\beta \, \mathbb{D}_{\text{KL}}\big[f_{\boldsymbol{\theta}} \,\|\, f_{\text{ref}}\big]_{i,t}}_{\text{KL penalty}}.$$

Using $u_{i,t} \triangleq \frac{f_{\text{ref}}(o_{i,t}|q,o_{i,<t})}{f_{\boldsymbol{\theta}}(o_{i,t}|q,o_{i,<t})} = \exp(\ell_{\text{ref}}(o_{i,t}) - \ell_{\boldsymbol{\theta}}(o_{i,t}))$,

$$\nabla_{\boldsymbol{\theta}} \, \mathbb{D}_{\text{KL}}[f_{\boldsymbol{\theta}}\|f_{\text{ref}}]_{i,t} = (1 - u_{i,t}) \, \nabla_{\boldsymbol{\theta}}\ell_{\boldsymbol{\theta}}(o_{i,t}).$$

The surrogate term contributes only when the unclipped branch is active (and is zero otherwise):

$$\nabla_{\boldsymbol{\theta}} \, \min(\cdot) = \mathbf{1}_{\text{unclipped}} \, \hat{A}_i \, \rho_{i,t} \, \nabla_{\boldsymbol{\theta}}\ell_{\boldsymbol{\theta}}(o_{i,t}),$$

**Gradient (batch-level).** Averaging over trajectories and tokens gives the batch-level gradient $\nabla_{\boldsymbol{\theta}}\mathcal{J}_{\text{GRPO}}(f_{\boldsymbol{\theta}})$:

$$\frac{1}{G} \sum_{i=1}^{G} \frac{1}{|o_i|} \sum_{t=1}^{|o_i|} \Big[\mathbf{1}_{\text{unclipped}} \, \hat{A}_i \, \rho_{i,t} - \beta\big(1-u_{i,t}\big)\Big] \nabla_{\boldsymbol{\theta}}\ell_{\boldsymbol{\theta}}(o_{i,t}),$$

where the unclipped indicator $\mathbf{1}_{\text{unclipped}}$ is

$$\mathbb{I}\Big[(\hat{A}_i \geq 0 \, \wedge \, \rho_{i,t} \leq 1 + \epsilon) \vee (\hat{A}_i < 0 \, \wedge \, \rho_{i,t} \geq 1 - \epsilon)\Big].$$

## 3.2. Empirical Findings

To characterize GRPO training dynamics, we post-train Qwen2.5-Math-1.5B (Yang et al., 2024b) with GRPO on MATH (Hendrycks et al., 2021b) and evaluate on MATH500 (Lightman et al., 2024). We define trajectory confidence as the mean token log-probability and question difficulty as the group error rate:

$$\text{Confidence}(f_{\boldsymbol{\theta}}, q, o_i) \triangleq \frac{1}{|o_i|} \sum_{t=1}^{|o_i|} \log f_{\boldsymbol{\theta}}(o_{i,t} \mid q, o_{i,<t}),$$

$$\text{Difficulty}(f_{\boldsymbol{\theta}}, q, a) \triangleq 1 - \frac{1}{G} \sum_{i=1}^{G} r_i.$$

We stratify trajectories into five difficulty bins: $[0, 0.2), [0.2, 0.4), [0.4, 0.6), [0.6, 0.8), [0.8, 1]$. Then, we report the corresponding training dynamics across these bins in Figs. 2-3, leading to the following observations. [1]

- **Confidence inflation.** During training, model confidences concentrate near 1 for both correct and incorrect outputs, consistent with entropy collapse and worsening calibration between confidence and accuracy (Yu et al., 2025).

- **Advantage contraction.** As accuracy increases, group-normalized advantages collapse toward 0: most samples have small nonnegative advantages, while incorrect samples become rarer but carry large negative values.

- **Hierarchical convergence.** Easy questions begin with high confidence and large gradients and quickly saturate as advantages/gradients vanish. Hard questions improve from low-confidence, low-gradient regimes, yet gradients decay rapidly, and a nontrivial error mass persists.

---

[1]Please see Appendix B.1 & B.2 for more results and analysis.

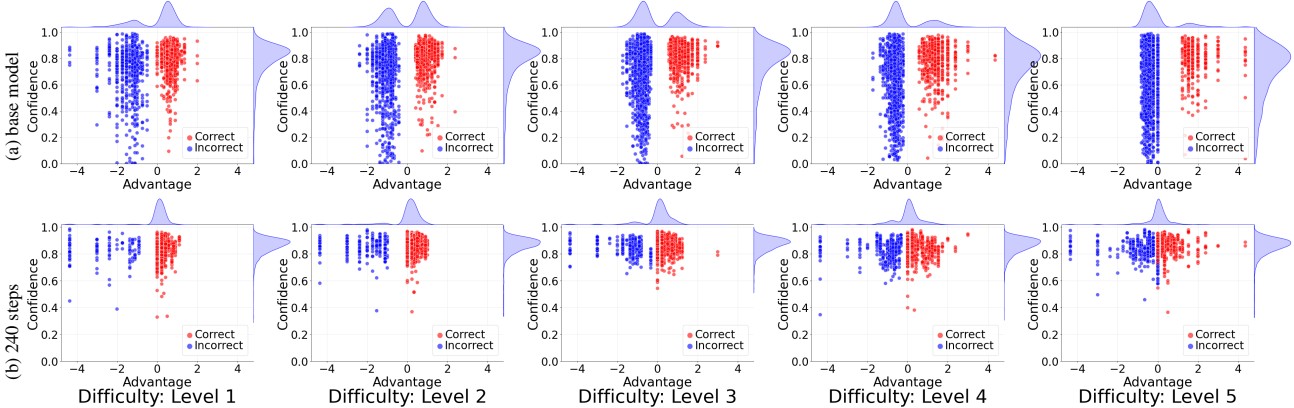

*Figure 2.* The confidence-advantage distribution of correct/incorrect trajectories in GRPO training. Rows 1 and 2 present the model checkpoints captured at training steps 0 (base model) and 240 (post-trained). Columns 1 to 5 correspond to difficulty levels of questions.

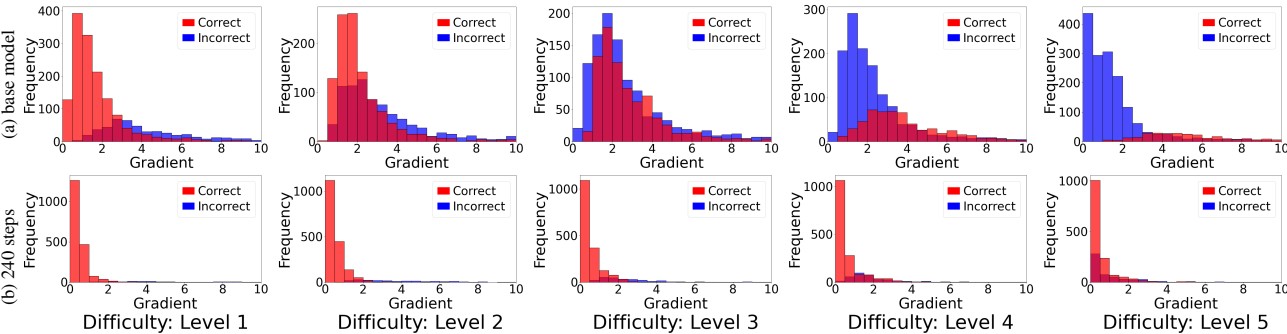

*Figure 3.* The gradient distribution of correct/incorrect trajectories (the same trajectories as in Fig. 2).

### 3.3. Mechanism Analysis

We give a compact mathematical account of the three observations. Specifically, we trace confidence inflation to the asymmetric clipping mechanism, advantage contraction to the structure of group-normalized advantages under improving accuracy, and hierarchical convergence to the discovery-limited nature of hard questions under finite sampling. We provide the full derivations in Appendix B.3-B.5.

**Why probability increases (and entropy collapses)?** A first-order ascent step on $\ell_{\boldsymbol{\theta}}(o_{i,t})$ has weight

$$\Delta\ell_{\boldsymbol{\theta}}(o_{i,t}) \propto w_{i,t} \triangleq \mathbf{1}_{\text{unclipped}} \; \hat{A}_i \; \rho_{i,t} - \beta\,(1 - u_{i,t}).$$

Clipping induces an asymmetric effect:

$$\hat{A}_i > 0: \; \mathbf{1}_{\text{unclipped}} = 1 \; \text{ while } \rho_{i,t} \le 1 + \epsilon,$$
$$\hat{A}_i < 0: \; \mathbf{1}_{\text{unclipped}} = 0 \; \text{ once } \rho_{i,t} < 1 - \epsilon.$$

Thus, positive-advantage tokens continue receiving upward pressure until the upper clipping threshold is reached, whereas negative-advantage tokens stop receiving downward pressure once they cross the lower threshold. Meanwhile, the KL term is restorative, acting mainly as a floor that pulls reduced probabilities back toward $f_{\text{ref}}$. Because the same trajectory-level advantage $\hat{A}_i$ is applied to all tokens in

a rollout, repeated positive updates concentrate probability mass on sampled tokens, thereby increasing confidence and reducing entropy.

**Why advantage contracts?** Let $\bar{r} = \frac{1}{G} \sum_{i=1}^{G} r_i$. Then

$$\hat{A}_i = \frac{r_i - \bar{r}}{\sqrt{\bar{r}(1 - \bar{r}) + \delta}}.$$

Ignoring $\delta$, within a group $\hat{A}_i \in \{\hat{A}^{(+)}, \hat{A}^{(-)}\}$ where

$$\hat{A}^{(+)}(\bar{r}) = \sqrt{\frac{1 - \bar{r}}{\bar{r}}}, \qquad \hat{A}^{(-)}(\bar{r}) = -\sqrt{\frac{\bar{r}}{1 - \bar{r}}}. \quad (1)$$

As $\bar{r} \uparrow 1$, $\hat{A}^{(+)}(\bar{r}) \downarrow 0$ while $\hat{A}^{(-)}(\bar{r}) \to -\infty$. Pooling across groups gives the two-atom mixture

$$\hat{A} \mid \bar{r} \sim \bar{r}\,\delta_{\hat{A}^{(+)}(\bar{r})} + (1 - \bar{r})\,\delta_{\hat{A}^{(-)}(\bar{r})},$$

so mass concentrates near 0 since $\bar{r} \to 1$ and $\hat{A}^{(+)} \to 0$, while the negative tail becomes increasingly rare.

**Why training converges hierarchically?** For question $q$, let $\pi(q) \triangleq \mathbb{P}_{o \sim f_{\text{old}}(\cdot|q)}[r(o) = 1]$ be the per-sample success rate. With $G$ samples and small $\pi(q)$, the probability of observing at least one correct trajectory is

$$\mathbb{P}(\exists\, i : r_i = 1 \mid q) = 1 - (1 - \pi(q))^G \approx G\pi(q). \quad (2)$$

Easy questions have large $\pi(q)$, hence frequent discovery and rapid reinforcement up to $\rho \approx 1 + \epsilon$; as $\bar{r} \uparrow 1$, Eq. 1 gives $\hat{A}^{(+)} \downarrow 0$, annealing gradients to zero. Hard questions have small $\pi(q)$, so learning is discovery-limited by Eq. 2, and each discovered success is only *capped*-amplified by clipping, yielding slow improvement and persistent error.

We note that the independence assumption on $\pi(q)$ is a simplification, while the exact within-group correlation is generally intractable and depends on factors such as output distribution, temperature, and prompting strategy. In practice, this assumption is optimistic: within-group correlation causes rollouts to fail in similar ways, making the true discovery probability lower than what Eq. 2 suggests.

# 4. Confidence and Difficulty-adaptive Policy Optimization (CoDaPO)

Existing RL methods typically optimize sampled trajectories with nearly uniform weighting, even though training utility depends on both question difficulty and the model's current competence. As shown in Sec. 3, this can overemphasize easy questions and lead to inefficient compute allocation.

Motivated by these findings, we propose CoDaPO, a *data-centric* and *model-adaptive* framework that plugs into standard RL objectives by reweighting policy updates and directing sampling toward more informative questions. Rather than resolving overconfidence or raising the theoretical ceiling of RL post-training, CoDaPO provides a stable and compute-efficient procedure for improving reasoning accuracy under a fixed training budget. Concretely, CoDaPO estimates a per-question value from confidence and difficulty (**CoDaWeighting**), resamples questions according to these values (**CoDaSampling**), and optimizes the policy with a value-weighted objective (**CoDaLearning**). Next, we present the overall framework in Sec. 4.1, implementation details in Sec. 4.2, and mechanism analysis in Sec. 4.3.

## 4.1. Training Framework

At each training step, we sample a mini-batch $\mathcal{B} = \{(q^{(j)}, a^{(j)})\}_{j=1}^{B} \sim \mathcal{D}$. For each question $q^{(j)}$, we use the behavior policy $f_{\text{old}}$ to generate a group of $G$ rollouts

$$\mathcal{O}_{\mathcal{B}} \triangleq \left\{\{o_i^{(j)}\}_{i=1}^{G}\right\}_{j=1}^{B}, \ \{o_i^{(j)}\}_{i=1}^{G} \sim f_{\text{old}}(\cdot \mid q^{(j)}).$$

**CoDaWeighting (per-question value estimation).** Given rollouts $\{o_i^{(j)}\}_{i=1}^{G}$, we assign each question a scalar value

$$v^{(j)} \triangleq \texttt{CoDaWeighting}\left(q^{(j)}, a^{(j)}, \{o_i^{(j)}\}_{i=1}^{G}\right).$$

The value $v^{(j)}$ reflects how informative the question is for further optimization (e.g., due to high difficulty or under-training). We collect the values as $\mathcal{V}_{\mathcal{B}} \triangleq \{v^{(j)}\}_{j=1}^{B}$.

**CoDaSampling (value-guided subset selection).** Using $\mathcal{V}_{\mathcal{B}}$, we form a value-biased resampled batch $\mathcal{S} \subseteq \mathcal{B}$ with $|\mathcal{S}| = |\mathcal{B}| = B$ (sample top-$K$ questions with replacement):

$$\mathcal{S} \triangleq \texttt{CoDaSampling}(\mathcal{B}, \mathcal{V}_{\mathcal{B}}, K).$$

We then resample $G$ trajectories for the selected questions to obtain fresh rollouts for learning:

$$\mathcal{O}_{\mathcal{S}} \triangleq \left\{\{o_i^{(j)}\}_{i=1}^{G}\right\}_{(q^{(j)}, a^{(j)}) \in \mathcal{S}}, \ \{o_i^{(j)}\}_{i=1}^{G} \sim f_{\text{old}}(\cdot \mid q^{(j)}).$$

The corresponding values $\mathcal{V}_{\mathcal{S}} \triangleq \{v^{(j)} : (q^{(j)}, a^{(j)}) \in \mathcal{S}\}$.

**CoDaLearning (two-stage policy update).** Finally, we update the current policy with a *batch-wide* step followed by a *focused* step on $\mathcal{S}$: [2]

$$f_{\boldsymbol{\theta}} \leftarrow \texttt{CoDaLearning}(f_{\boldsymbol{\theta}}, \mathcal{B}, \mathcal{O}_{\mathcal{B}}, \mathcal{V}_{\mathcal{B}}),$$
$$f_{\boldsymbol{\theta}} \leftarrow \texttt{CoDaLearning}(f_{\boldsymbol{\theta}}, \mathcal{S}, \mathcal{O}_{\mathcal{S}}, \mathcal{V}_{\mathcal{S}}).$$

The first update preserves broad coverage over the batch, while the second reallocates computation toward high-value questions. The full pipeline is summarized in Algorithm 1.

---

**Algorithm 1** The training pipeline of CoDaPO

---

**Input:** Initial policy model $f_{\boldsymbol{\theta}}$, training set $\mathcal{D}$, batch size $B$, group size $G$, sample size $K$

1: **for** step $= 1, \cdots, M$ **do**
2:      Sample a mini-batch $\mathcal{B} = \{(q^{(j)}, a^{(j)})\}_{j=1}^{B} \sim \mathcal{D}$
3:      Update the behavior policy: $f_{\text{old}} \leftarrow f_{\boldsymbol{\theta}}$
4:      // CoDaWeighting: rollout collection and value estimation
5:      **for** each $(q^{(j)}, a^{(j)}) \in \mathcal{B}$ **do**
6:         Sample $\{o_i^{(j)}\}_{i=1}^{G} \sim f_{\text{old}}(\cdot \mid q^{(j)})$
7:         Set $v^{(j)} \leftarrow \texttt{CoDaWeighting}\left(q^{(j)}, a^{(j)}, \{o_i^{(j)}\}_{i=1}^{G}\right)$
8:      **end for**
9:      Set $\mathcal{V}_{\mathcal{B}} \leftarrow \{v^{(j)}\}_{j=1}^{B}$ and $\mathcal{O}_{\mathcal{B}} \leftarrow \{\{o_i^{(j)}\}_{i=1}^{G}\}_{j=1}^{B}$
10:      // CoDaSampling: value-guided resampling
11:      Sample $\mathcal{S} \leftarrow \texttt{CoDaSampling}(\mathcal{B}, \mathcal{V}_{\mathcal{B}}, K)$
12:      // Fresh rollouts for the resampled batch
13:      **for** each $(q^{(j)}, a^{(j)}) \in \mathcal{S}$ **do**
14:         Sample $\{o_i^{(j)}\}_{i=1}^{G} \sim f_{\text{old}}(\cdot \mid q^{(j)})$
15:         Set $v^{(j)} \leftarrow \texttt{CoDaWeighting}\left(q^{(j)}, a^{(j)}, \{o_i^{(j)}\}_{i=1}^{G}\right)$
16:      **end for**
17:      Set $\mathcal{V}_{\mathcal{S}} \leftarrow \{v^{(j)} : (q^{(j)}, a^{(j)}) \in \mathcal{S}\}$
18:      Set $\mathcal{O}_{\mathcal{S}} \leftarrow \{\{o_i^{(j)}\}_{i=1}^{G}\}_{(q^{(j)}, a^{(j)}) \in \mathcal{S}}$
19:      // CoDaLearning: two-stage policy update
20:      Update $f_{\boldsymbol{\theta}} \leftarrow \texttt{CoDaLearning}(f_{\boldsymbol{\theta}}, \mathcal{B}, \mathcal{O}_{\mathcal{B}}, \mathcal{V}_{\mathcal{B}})$
21:      Update $f_{\boldsymbol{\theta}} \leftarrow \texttt{CoDaLearning}(f_{\boldsymbol{\theta}}, \mathcal{S}, \mathcal{O}_{\mathcal{S}}, \mathcal{V}_{\mathcal{S}})$
22: **end for**

**Output:** Updated policy model $f_{\boldsymbol{\theta}}$

---

[2]CoDaPO reallocates compute within a *fixed* computation budget rather than increasing the budget. We use 50% budget to generate/learn on mini-batch $\mathcal{B}$, and 50% on resampled-batch $\mathcal{S}$.

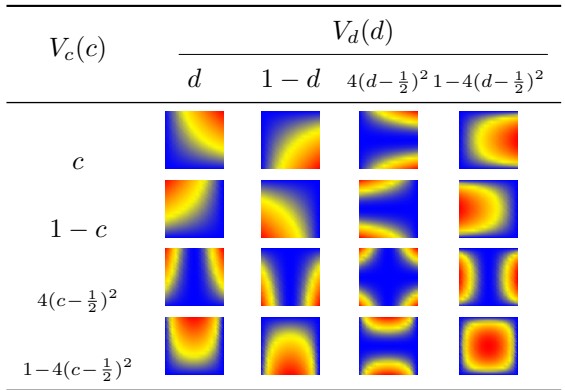

| $V_c(c)$ | $V_d(d)$ | | | |
|---|---|---|---|---|
| | $d$ | $1-d$ | $4(d-\frac{1}{2})^2$ | $1-4(d-\frac{1}{2})^2$ |
| $c$ | | | | |
| $1-c$ | | | | |
| $4(c-\frac{1}{2})^2$ | | | | |
| $1-4(c-\frac{1}{2})^2$ | | | | |

*Table 1.* Design choices for $V_c(\cdot)$ and $V_d(\cdot)$. Each cell visualizes the value function $V(c,d) = V_c(c)V_d(d)$ over $(c,d) \in [0,1]^2$.

### 4.2. Implementation

In this part, we detail the implementation of CoDaPO, organized into the following three components.

**CoDaWeighting.** Given a question $q$ and its rollout group $\{o_i\}_{i=1}^G$, we estimate the group confidence $c_q$ and difficulty $d_q$ as (by construction, $c_q \in (0,1]$ and $d_q \in [0,1)$)

$$c_q \triangleq \exp\left[\frac{1}{G}\sum_{i=1}^{G}\frac{1}{|o_i|}\sum_{t=1}^{|o_i|}\log f_{\boldsymbol\theta}(o_{i,t} \mid q, o_{i,<t})\right], \quad (3)$$

$$d_q \triangleq 1 - \frac{1}{G}\sum_{i=1}^{G} r_i. \quad (4)$$

We then map $(c_q, d_q)$ to a scalar value $v_q$ via two separable weighting functions, i.e., $v_q = V(c_q, d_q) = V_c(c_q)\,V_d(d_q)$. The design choices for $V_c(\cdot)/V_d(\cdot)$ are shown in Tab. 1. Empirically, we choose a linear $V_c(x) = x$ to encourage larger updates on questions the model is already confident about, and a U-shape $V_d(x) = 1 - 4(x - 1/2)^2$ to emphasize the "learnable" mid-difficulty regime while down-weighting nearly-solved and discovery-limited questions, yielding

$$v_q = V(c_q, d_q) = c_q\Big(1 - 4(d_q - 1/2)^2\Big). \quad (5)$$

**CoDaSampling.** Given the per-sample values $\mathcal{V}_\mathcal{B} \triangleq \{v^{(j)}\}_{j=1}^B$, we rank the batch $\mathcal{B}$ by $v^{(j)}$ and retain the top-$K$ question-answer pairs. We then form a resampled batch $\mathcal{S}$ by sampling *with replacement* from these top-$K$ pairs, repeating each selected pair $B/K$ times so that $|\mathcal{S}| = B$.

**CoDaLearning.** Given $(f_{\boldsymbol\theta}, \mathcal{B}, \mathcal{O}_\mathcal{B}, \mathcal{V}_\mathcal{B})$, CoDaPO updates the policy by maximizing a value-weighted GRPO objective; the same objective is applied to $(f_{\boldsymbol\theta}, \mathcal{S}, \mathcal{O}_\mathcal{S}, \mathcal{V}_\mathcal{S})$.

$$\mathcal{J}_{\text{CoDaPO}}(f_{\boldsymbol\theta}, \mathcal{B}, \mathcal{O}_\mathcal{B}, \mathcal{V}_\mathcal{B}) \triangleq \sum_{j=1}^{B}\frac{1}{\sum_{i=1}^{G}|o_i^{(j)}|}\sum_{i=1}^{G}\sum_{t=1}^{|o_i^{(j)}|}$$
$$\left[\min\Big(\rho_{i,t}^{(j)}\hat{A}_i^{(j)}, \text{clip}(\rho_{i,t}^{(j)}, 1-\epsilon, 1+\epsilon)\hat{A}_i^{(j)}\Big)v^{(j)}\right], \quad (6)$$

where $\rho_{i,t}^{(j)} \triangleq \frac{f_{\boldsymbol\theta}(o_{i,t}^{(j)}|q^{(j)}, o_{i,<t}^{(j)})}{f_{\text{old}}(o_{i,t}^{(j)}|q^{(j)}, o_{i,<t}^{(j)})}$ (the token-level importance ratio), and the group-normalized advantage is computed from binary rewards $r_i^{(j)}$ as $\hat{A}_i^{(j)} \triangleq \frac{r_i^{(j)} - \text{mean}(\{r_\ell^{(j)}\}_{\ell=1}^G)}{\text{std}(\{r_\ell^{(j)}\}_{\ell=1}^G)}$. Beyond value reweighting, we make two changes to GRPO, as in Yu et al. (2025); Chu et al. (2026):

- **Token-level micro-averaging.** We normalize the objective by the total number of tokens, $\frac{1}{\sum_i|o_i|}\sum_{i,t}(\cdot)$, instead of averaging per trajectory, $\frac{1}{G}\sum_i\frac{1}{|o_i|}\sum_t(\cdot)$. This makes each token contribute equally and removes the implicit length penalty that down-weights longer rollouts.

- **No KL regularization.** We drop the KL term to encourage exploration and avoid an additional $f_{\text{ref}}$ forward pass.

### 4.3. Mechanism Analysis

Here, we show that CoDaPO modifies GRPO by reshaping *effective token weights* and reallocating *compute* across questions. Full analysis is provided in Appendix C.

**CoDaPO optimizes a value-scaled token-level policy gradient.** With stop-gradient through $\hat{A}_i^{(j)}$ and $v^{(j)}$, the update takes the token-micro-averaged form

$$\nabla_{\boldsymbol\theta}\mathcal{J}_{\text{CoDaPO}} \propto \sum_{j=1}^{B}\frac{1}{\sum_i|o_i^{(j)}|}\sum_{i,t} w_{i,t}^{(j)}\,\nabla_{\boldsymbol\theta}\ell_{\boldsymbol\theta}(o_{i,t}^{(j)}),$$

where $w_{i,t}^{(j)} = v^{(j)}\,\mathbf{1}_{\text{unclipped}}\,\rho_{i,t}^{(j)}\,\hat{A}_i^{(j)}$. Thus, CoDaPO acts through $w_{i,t}^{(j)}$: it rescales GRPO's per-token ascent direction by a bounded, question-level factor $v^{(j)}$, while micro-averaging removes length as an implicit reweighting signal.

**CoDaWeighting concentrates gradients on "learnable" questions and suppresses uninformative updates.** From the rollout group of question $q$, we compute $(c_q, d_q)$ and assign $v_q = c_q(1 - 4(d_q - 1/2)^2)$. This implements a *learnable-band prior*: $v_q \approx 0$ when $d_q \approx 0$ (already solved, where updates mainly inflate confidence) or $d_q \approx 1$ (discovery-limited, where gradients are dominated by clipped negatives), and $v_q$ peaks near $d_q \approx \frac{1}{2}$ where correct trajectories occur often enough to yield actionable signal. On the other hand, $c_q$ measures how confident the model is in its own reasoning trajectory: whether it has committed to a coherent path or is producing uncertain, diffuse outputs. It works jointly with $d_q$ to identify questions in the "learnable band" for adaptive compute allocation.

**CoDaSampling boosts hard-case progress by increasing discovery probability via repeated trials.** Let $\pi(q) \triangleq \mathbb{P}_{o \sim f_{\text{old}}(\cdot|q)}[r(o) = 1]$ denote the per-rollout success probability. With group size $G$, the probability of observing at least one correct rollout is $p_{\text{disc}}(q) = 1 - (1 - \pi(q))^G$. If CoDaSampling repeats the same question $m$ times and

| Base Model | Algorithm | Datasets | | | | | | | Average |
|---|---|---|---|---|---|---|---|---|---|
| | | MATH 500 | AIME 2024 | AIME 2025 | AMC 2023 | Olympiad Bench | Minerva | GSM8K | |
| Llama-3.2-1B-Instruct | Base | 13.18 | 0.69 | 0.00 | 6.71 | 3.92 | 1.64 | 2.02 | 4.02 |
| | GRPO | 23.76 | 2.31 | 0.00 | 11.61 | **6.40** | 4.06 | 46.23 | 13.48 |
| | DAPO | 25.45 | 2.84 | 0.10 | 10.80 | 6.00 | **4.94** | 50.69 | 14.40 |
| | Dr. GRPO | 24.61 | 2.01 | 0.00 | 9.76 | 4.98 | 4.69 | **51.52** | 13.94 |
| | GPG | 23.03 | 1.16 | 0.00 | 11.86 | 4.91 | 4.62 | 45.42 | 13.00 |
| | **CoDaPO (ours)** | **27.39** | **3.18** | **0.39** | **11.97** | 6.36 | 4.86 | 51.20 | **15.05** |
| Qwen2.5-Math-1.5B | Base | 30.63 | 5.71 | 2.50 | 23.40 | 18.78 | 5.29 | 29.57 | 16.55 |
| | GRPO | 70.31 | 13.02 | 8.00 | 50.84 | 32.18 | 16.37 | 82.86 | 39.08 |
| | DAPO | 70.02 | 13.15 | 12.20 | 50.35 | 32.87 | 17.85 | 80.00 | 39.49 |
| | Dr. GRPO | 68.35 | 12.70 | 8.15 | 50.61 | 31.39 | 16.69 | 82.56 | 38.64 |
| | GPG | 69.89 | **14.63** | 8.03 | 51.62 | 32.72 | 17.98 | 83.51 | 39.77 |
| | **CoDaPO (ours)** | **71.54** | 14.47 | **12.35** | **52.68** | **36.16** | **18.04** | **83.86** | **41.30** |
| Qwen2.5-Math-7B | Base | 54.00 | 16.37 | 6.75 | 51.07 | 27.53 | 13.28 | 61.56 | 32.94 |
| | GRPO | 72.18 | 27.40 | 11.07 | 62.19 | 37.35 | 18.55 | 83.35 | 44.58 |
| | DAPO | 73.13 | 29.77 | 10.06 | 59.16 | 36.18 | 20.00 | 86.94 | 45.03 |
| | Dr. GRPO | 72.37 | 25.04 | 10.11 | 62.06 | 36.00 | 21.09 | 85.93 | 44.66 |
| | GPG | 72.57 | 27.23 | **12.98** | 61.99 | 36.26 | 21.06 | 81.24 | 44.76 |
| | **CoDaPO (ours)** | **74.39** | **30.49** | 11.46 | **63.50** | **37.98** | **21.63** | **87.21** | **46.67** |

*Table 2.* The main results of the post-training experiments (accuracy, in %). Note that the **boldface** numbers mean the best results, while the underlined numbers indicate the second-best results.

draws fresh groups, the probability of *ever* seeing a correct rollout becomes $1 - (1 - p_{\text{disc}}(q))^m = 1 - \left(1 - \pi(q)\right)^{Gm}$.

Here, the repetition by CoDaSampling alleviates the discovery bottleneck that drives hierarchical convergence: when $\pi(q)$ is small, correct rollouts are rarely observed, so learning stalls. By increasing the probability of observing at least one correct trajectory, CoDaSampling more reliably triggers the subsequent *amplification* phase, where correct rollouts obtain positive advantages and are reinforced.

**Overall effect: fewer saturated updates, faster discovery, and implicit annealing on easy questions.** CoDaPO does not alter clipping asymmetry (Sec. 3), but it reduces the *volume* of uninformative saturated updates by deallocating solved questions ($d_q \downarrow 0 \Rightarrow v_q \downarrow 0$) and by concentrating repeated trials where discovery is plausible. Meanwhile, advantage contraction becomes a feature rather than a bottleneck: as $\bar{r} \uparrow 1$, $\hat{A}^{(+)}(\bar{r}) \downarrow 0$ and $d_q \downarrow 0$, so both $\hat{A}$ and $v_q$ shrink, annealing gradients on easy $q$ and preserving capacity for harder $q$ within the same compute budget.

## 5. Experiments

### 5.1. Setup

**Training setup.** We post-train Llama-3.2-1B-Instruct, Qwen2.5-Math-1.5B, and Qwen2.5-Math-7B (Grattafiori et al., 2024; Yang et al., 2024b) on MATH (Hendrycks et al., 2021b), with AlphaApollo (Zhou et al., 2025b) and 4×A100 GPUs. We set the batch size to 16 and sample 8 rollouts per group for each question. To ensure a fair comparison, we count the additional training steps introduced by CoDaPO

toward the total training steps. All experiments share the same total training step budget.

**Evaluation setup.** We adopt the Qwen2.5-Math evaluation codebase for consistent measurement. For each question, we sample 32 responses at temperature 0.6 and report the mean accuracy. Evaluations span seven mathematical reasoning benchmarks: MATH500 (Lightman et al., 2024; Hendrycks et al., 2021b), AIME 2024, AIME 2025, AMC 2023, OlympiadBench (He et al., 2024), Minerva (Lewkowycz et al., 2022), and GSM8K (Cobbe et al., 2021). For broader reasoning capabilities, we additionally evaluate on science benchmarks including MMLU (Hendrycks et al., 2021a) and GPQA (Rein et al., 2024), as well as coding benchmarks including HumanEval (Chen et al., 2021), TACO (Li et al., 2023a), and LiveCodeBench (Jain et al., 2025).

**Baselines.** We compare CoDaPO against representative RL baselines, including GRPO (Shao et al., 2024) and several recent concurrent methods: DAPO (Yu et al., 2025), Dr. GRPO (Liu et al., 2025a), and GPG (Chu et al., 2026). All baselines are implemented under the same training and evaluation settings for fair comparison.

### 5.2. Main results

**Mathematical reasoning performance.** *CoDaPO consistently improves mathematical reasoning performance across all evaluated models and benchmarks.* As shown in Tab. 2, on Qwen2.5-Math-1.5B, CoDaPO raises the average accuracy over seven benchmarks from 16.55% to 41.30% and achieves the best results on most datasets. CoDaPO also scales effectively to larger models: when applied to

| Model | MMLU | GPQA | HumanEval | Average |
|---|---|---|---|---|
| Qwen2.5-Math-1.5B | 11.53 | 9.85 | 29.27 | 16.88 |
| GRPO | 43.46 | 19.69 | 34.76 | 32.64 |
| CoDaPO | **44.81** | **24.45** | **50.61** | **39.96** |

Table 3. Evaluation results on cross-domain benchmarks.

| Pass@K | 1 | 2 | 4 | 8 | 16 | 32 | 64 | 128 |
|---|---|---|---|---|---|---|---|---|
| Qwen2.5-Math-1.5B | 3.33 | 3.33 | 6.67 | 13.33 | 16.67 | 16.67 | 26.67 | 46.67 |
| GRPO | 6.67 | 13.33 | 20.00 | 20.00 | 26.67 | 36.67 | 43.33 | 46.67 |
| CoDaPO | **13.33** | **20.00** | **23.33** | **30.00** | **36.67** | **40.00** | **46.67** | **53.33** |

Table 4. Pass@K result of different methods on AIME25.

Qwen2.5-Math-7B under the same settings, it achieves the best average accuracy of $46.67\%$, demonstrating its robustness and consistent effectiveness across model sizes. Moreover, CoDaPO consistently outperforms all RL baselines, achieving relative accuracy gains of $5.68\%$ over GRPO and surpassing other strong baselines on Qwen2.5-Math-1.5B.

Importantly, these gains are not limited to a specific backbone. When applied to Llama-3.2-1B-Instruct, CoDaPO again achieves the best overall performance among all methods, demonstrating that its effectiveness generalizes beyond the Qwen2.5-Math family and does not rely on a particular model backbone. Notably, although post-trained only on MATH, CoDaPO generalizes well to diverse mathematical benchmarks, with substantial gains on challenging out-of-domain sets such as OlympiadBench and Minerva, indicating enhanced transferable reasoning ability.

**Cross-domain generalization.** We evaluate CoDaPO on science and coding benchmarks to assess its generalization beyond mathematics. As shown in Tab. 3, CoDaPO consistently surpasses GRPO across all evaluated tasks, achieving higher performance on MMLU (Hendrycks et al., 2021a), GPQA (Rein et al., 2024), and HumanEval (Chen et al., 2021). These results suggest that *CoDaPO improves general reasoning behaviors that transfer beyond the math domain.*

**Test-time scaling.** As shown in Tab. 4, *CoDaPO consistently outperforms GRPO across all values of K, demonstrating superior sample efficiency and more effective utilization of additional compute.* In particular, under small-sample regimes, CoDaPO gains improvements of up to $10.00\%$ over GRPO. As $K$ increases, CoDaPO continues to perform well and reaches a Pass@128 of $53.33\%$.

### 5.3. Ablation studies

**Individual components.** We first ablate the three components in CoDaPO by progressively adding them on top of the same base model and training budget. Tab. 5 compares (i) GRPO, (ii) GRPO with only CoDaWeighting, and (iii) the full method with CoDaSampling enabled. Adding CoDaWeighting consistently improves over vanilla GRPO (Average: $30.44\% \rightarrow 31.53\%$), indicating that value-aware reweighting effectively suppresses uninformative updates

| Model | MATH500 | AIME2024 | AIME2025 | Average |
|---|---|---|---|---|
| Qwen2.5-Math-1.5B | 30.63 | 5.71 | 2.50 | 12.95 |
| +GRPO | 70.31 | 13.02 | 8.00 | 30.44 |
| +CoDaWeighting | 71.09 | 13.90 | 9.59 | 31.53 |
| +CoDaSampling | **71.54** | **14.47** | **12.35** | **32.79** |

Table 5. Ablation of individual components in CoDaPO.

| $V_c(c)$ | $V_d(d)$ | | | |
|---|---|---|---|---|
| | $d$ | $1-d$ | $4(d-\frac{1}{2})^2$ | $1-4(d-\frac{1}{2})^2$ |
| $c$ | 31.14 | 31.74 | 30.17 | **31.98** |
| $1-c$ | 30.15 | 30.34 | 29.99 | 31.16 |
| $4(c-\frac{1}{2})^2$ | 29.67 | 31.59 | 29.85 | 31.37 |
| $1-4(c-\frac{1}{2})^2$ | 30.23 | 31.48 | 29.74 | 31.59 |

Table 6. Ablation on CoDaWeighting designs ($V_c$ and $V_d$).

and reallocates gradient mass toward more learnable questions. Further enabling CoDaSampling yields an additional gain (Average: $31.53\% \rightarrow 32.79\%$), suggesting that repeatedly sampling high-value questions increases the discovery probability of correct rollouts and accelerates hard-case progress. *Overall, each component contributes non-trivially, and the best performance is achieved only when both CoDaWeighting and CoDaSampling are used together.*

**CoDaWeighting.** We then study different designs of the separable value function $V(c,d) = V_c(c) V_d(d)$ in CoDaWeighting. Tab. 6 enumerates representative choices of $V_c(\cdot)$ and $V_d(\cdot)$, covering monotonic and symmetric/U-shaped forms. We observe clear performance differences among weighting strategies: designs that tend to over-emphasize either very hard or very easy items can increase the fraction of saturated or discovery-limited updates and thus hurt overall efficiency. In contrast, the combination $V_c(c) = c$ and $V_d(d) = 1-4(d-\frac{1}{2})^2$ consistently performs best, consistent with our design goal of suppressing both very easy ($d\approx0$) and extremely hard ($d\approx1$) items, and emphasizing questions of intermediate difficulty that provide more informative gradients. Building on this observation, we further isolate the contribution of each factor by retaining only CoDaWeighting while setting either $V_c(c) = 1$ or $V_d(d) = 1$. The results are shown in Tab. 7. We find that both the confidence term $V_c(c)$ and the difficulty term $V_d(d)$ individually improve over the GRPO baseline, while combining the two yields the best overall performance (Average: $31.53\%$). *This suggests that the two factors provide complementary benefits: $V_c(c)$ helps prioritize more reliable learning signals, whereas $V_d(d)$ suppresses both saturated easy samples and unproductive overly difficult samples.*

**CoDaSampling.** Next, we vary the Top-$K$ hyperparameter in CoDaSampling, which controls how many high-value question-answer pairs are retained for resampling within each batch. As shown in Tab. 8, different $K \in \{2, 4, 8\}$ lead to broadly similar results (Average: $32.48\%/32.79\%/32.62\%$), indicating that CoDaSampling is not overly sensitive to this choice. *Extremely small K*

| $V_c(c)$ | $V_d(d)$ | MATH500 | AIME24 | AIME25 | Avg |
|---|---|---|---|---|---|
| 1 | $1 - 4(d - \frac{1}{2})^2$ | 70.65 | **14.29** | 9.02 | 31.32 |
| $c$ | 1 | 70.60 | 13.11 | 9.26 | 30.99 |
| $c$ | $1 - 4(d - \frac{1}{2})^2$ | **71.09** | 13.90 | **9.59** | **31.53** |
| GRPO | — | 70.31 | 13.02 | 8.00 | 30.44 |

*Table 7.* Ablation study on $V_c(c)$ and $V_d(d)$ in CoDaWeighting.

| Top-K | MATH500 | AIME2024 | AIME2025 | Average |
|---|---|---|---|---|
| 1 | 69.82 | 13.04 | 8.33 | 30.40 |
| 2 | 70.19 | **16.41** | 10.83 | 32.48 |
| 4 | **71.54** | 14.47 | **12.35** | **32.79** |
| 8 | 71.10 | 15.71 | 11.04 | 32.62 |

*Table 8.* Ablation on the Top-$K$ choice in CoDaSampling.

| Algorithm | MATH 500 | AIME 2024 | AIME 2025 | AMC 2023 | Olympiad Bench | Minerva | GSM8K | Avg |
|---|---|---|---|---|---|---|---|---|
| DAPO | 70.02 | 13.15 | **12.20** | 50.35 | 32.87 | **17.85** | 80.00 | 39.49 |
| +CoDaPO | **70.74** | **17.00** | 10.27 | **53.00** | **34.11** | 17.84 | **84.25** | **41.03** |
| GPG | 69.89 | **14.63** | 8.03 | 51.62 | **32.72** | 17.98 | 83.51 | 39.77 |
| +CoDaPO | **70.15** | 14.36 | **8.66** | **52.38** | 32.31 | **18.12** | **83.66** | **39.95** |
| GRPO | 70.31 | 13.02 | 8.00 | 50.84 | 32.18 | 16.37 | 82.86 | 39.08 |
| +CoDaPO | **71.54** | **14.47** | **12.35** | **52.68** | **36.16** | **18.04** | **83.86** | **41.30** |

*Table 9.* CoDaLearning applied to different RL objectives.

*(e.g., $K = 1$) can be slightly less stable due to insufficient sample diversity, as the model focuses too heavily on a single question, while larger $K$ provides more diversity but weakens the guidance of CoDaWeighting.* We choose $K = 4$ as a simple default that balances focused compute allocation with within-batch diversity, and it achieves the best (or near-best) average performance.

**CoDaLearning.** Finally, we evaluate the generality of Co-DaLearning by applying the same value-weighted learning rule to different RL objectives. Tab. 9 compares DAPO, GPG, and GRPO with and without CoDaLearning under the same training setting. Across all three baselines, Co-DaLearning brings consistent gains on most benchmarks (Avg: DAPO 39.49% → 41.03%; GPG 39.77% → 39.95%; GRPO 39.08% → 41.30%), indicating that *the same value weighting can consistently improve multiple RL objectives.*

**Compared with curriculum learning.** We compare Co-DaPO with curriculum strategies that organize training data according to pre-defined difficulty schedules. As shown in Tab. 10, a standard easy→hard curriculum only slightly improves GRPO from 39.08% to 39.30% average accuracy, whereas CoDaPO achieves a substantially larger gain to 41.30%. Importantly, CoDaPO is not a curriculum method, but a dynamics-driven compute allocation framework. Unlike monotonic curricula, CoDaPO adopts a non-monotonic weighting strategy that suppresses both very easy and extremely hard samples while emphasizing intermediate-difficulty questions that provide more informative gradients. Moreover, CoDaPO jointly incorporates both confidence

| Training Data | MATH 500 | AIME 2024 | AIME 2025 | AMC 2023 | Olympiad Bench | Minerva | GSM8K | Avg |
|---|---|---|---|---|---|---|---|---|
| GRPO (full MATH) | 70.31 | 13.02 | 8.00 | 50.84 | 32.18 | 16.37 | 82.86 | 39.08 |
| GRPO (Level-3 only) | 69.56 | 11.42 | 8.31 | 49.79 | 33.85 | 16.75 | 82.73 | 38.92 |
| GRPO (Level-5 only) | 68.01 | 13.47 | 7.77 | 46.53 | 30.94 | 17.20 | 81.71 | 37.95 |
| GRPO (Level 1→5 curriculum) | 70.19 | 14.39 | 11.55 | 48.95 | 32.22 | 17.05 | 80.77 | 39.30 |
| **CoDaPO (full MATH)** | **71.54** | **14.47** | **12.35** | **52.68** | **36.16** | **18.04** | **83.86** | **41.30** |

*Table 10.* Comparison between CoDaPO and curriculum strategies.

| Algorithm | TACO | HumanEval | LiveCodeBench | Avg |
|---|---|---|---|---|
| Base | 15.47 | 29.27 | 9.00 | 17.91 |
| GRPO | 50.92 | 50.61 | 48.75 | 50.09 |
| **CoDaPO** | **52.66** | **54.88** | **54.00** | **53.85** |

*Table 11.* Performance on coding tasks by pass rate.

and difficulty through $v_q = V_c(c_q)V_d(d_q)$ and further performs adaptive gradient reweighting and two-stage updates during optimization, *which cannot be replicated by static data scheduling alone.* The inferior results of training only on Level-3 (38.92%) or Level-5 (37.95%) subsets further suggest that simple difficulty filtering is insufficient.

**Generalization to coding tasks.** We further evaluate CoDaPO in the coding domain. Specifically, we train on the TACO (Li et al., 2023a) training set and evaluate on TACO, HumanEval (Chen et al., 2021), and Live-CodeBench (Jain et al., 2025) using pass rate as the metric. As shown in Tab. 11, CoDaPO consistently improves over GRPO across all benchmarks, increasing the average performance from 50.09% to 53.85%. Notably, CoDaPO achieves gains of +1.74% on TACO, +4.27% on HumanEval, and +5.25% on LiveCodeBench, demonstrating that *the benefits of dynamics-aware compute allocation generalize beyond mathematical reasoning to code generation tasks.*

**Case Studies.** We present several representative case studies in Appendix E. Compared to GRPO, CoDaPO demonstrates more coherent and structured reasoning trajectories.

## 6. Conclusion

In this work, we analyze GRPO's training dynamics for verifiable long-horizon reasoning and identify three recurring behaviors, confidence inflation, advantage contraction, and hierarchical convergence, that expose inefficient compute allocation under uniform sampling and weighting. Motivated by these insights, we propose CoDaPO, a data-centric RL framework that reweights policy updates and resamples high-value questions to increase discovery under a fixed budget. Experiments on twelve reasoning benchmarks across mathematics, science, and coding demonstrate that CoDaPO consistently outperforms RL baseline methods.

## Acknowledgements

ZKZ, XYL, CTC, and BH were supported by RGC Young Collaborative Research Grant No. C2005-24Y and NSFC Major Research Plan No. 92570109. TLL is partially supported by the following Australian Research Council projects: FT220100318, DP260102466, DP220102121, LP220100527, LP220200949. SK acknowledges support by NSF 2046795 and 2205329, IES R305C240046, ARPA-H, the MacArthur Foundation, Schmidt Sciences, and HAI. The authors also thank Jianghangfan Zhang for his assistance with the experiments.

## Impact Statement

This work aims to advance the field of machine learning and large language models, especially the capabilities of machine reasoning. There are many potential societal consequences of our work, none of which we feel must be specifically highlighted here. The study also does not involve human subjects, data set releases, potentially harmful insights, applications, conflicts of interest, sponsorship, discrimination, bias, fairness concerns, privacy or security issues, legal compliance issues, or research integrity issues.

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

## Appendix

# A. Related Work

**Supervised Fine-tuning (SFT)** finetunes the policy model to predict the next token on data that is more relevant to the downstream task. The objective of SFT is to maximize the token-wise log probability of dataset-collected outputs $\mathcal{O}$, which are treated as the ground truth for training. Namely,

$$\mathcal{J}_{\text{SFT}}(f_{\boldsymbol{\theta}}) \triangleq \mathbb{E}_{(q,a)\sim\mathcal{D},o\sim\mathcal{O}(q)} \left( \frac{1}{|o|} \sum_{t=1}^{|o|} \log f_{\boldsymbol{\theta}}(o_t|q, o_{<t}) \right).$$

Although simple in optimization, SFT has several significant drawbacks. SFT focuses on exploiting (memorizing, to some extent) the dataset-collected outputs, resulting in limited generalization power, especially in the out-of-distribution scenarios (Chu et al., 2025). Besides, collecting and annotating the output data can be expensive and often requires domain-specific knowledge in solving particular questions.

**Proximal Policy Optimization (PPO)** (Schulman et al., 2017) is an actor-critic RL algorithm that is widely used in the RL fine-tuning stage of LLMs. Simplifying the TRPO (Schulman et al., 2015), PPO maximizes the advantage $A_t$ of the model-generated output $o$ without the need to collect ground truth outputs. Here, the advantage $A_t$ is computed by the Generalized Advantage Estimation (GAE) (Schulman et al., 2016), taking 1) the output value estimated by a trainable value model and 2) the KL penalty between $f_{\boldsymbol{\theta}}$ and $f_{\text{ref}}$. PPO maximizes the objective:

$$\mathcal{J}_{\text{PPO}}(f_{\boldsymbol{\theta}}) \triangleq \mathbb{E}_{(q,a)\sim\mathcal{D},o\sim f_{\text{old}}(\cdot|q)} \frac{1}{|o|} \sum_{t=1}^{|o|} \min\left[ \frac{f_{\boldsymbol{\theta}}(o_t|q, o_{<t})}{f_{\text{old}}(o_t|q, o_{<t})} A_t, \text{clip}\left( \frac{f_{\boldsymbol{\theta}}(o_t|q, o_{<t})}{f_{\text{old}}(o_t|q, o_{<t})}, 1-\epsilon, 1+\epsilon \right) A_t \right].$$

Although widely used in alignment tasks, PPO has several limitations. Its learning process is unstable, computationally expensive, and requires extensive hyperparameter tuning. The clipped objective can slow convergence and yield suboptimal policies (Engstrom et al., 2020). Notably, training the value model is challenging due to high variance and poor generalization (OpenAI et al., 2018). Besides, PPO is also prone to reward hacking, struggles with long-term credit assignment, and suffers from the issue of sample inefficiency (Henderson et al., 2018).

**Group Relative Policy Optimization (GRPO)** (Shao et al., 2024) simplifies PPO via removing the learnable value model. Instead, GRPO uses the average reward of multiple sampled outputs for the same question. Specifically, given a question $q$, GRPO requires to sample $G$ outputs from the old policy as $\{o_i\}_{i=1}^{G} \sim f_{\text{old}}(\cdot|q)$. Then, it computes the reward $r_i$ for each output $o_i$ (through deterministic reward functions) and obtains a group of rewards $\{r_i\}_{i=1}^{G}$. The advantage $\hat{A}_i$ of GRPO is estimated as:

$$\hat{A}_i = \tilde{r}_i = \frac{r_i - \text{mean}(\{r_j\}_{j=1}^{G})}{\text{std}(\{r_j\}_{j=1}^{G})}.$$

The objective of GRPO, shown below, is to maximize the advantage (the first term) while ensuring that the policy model remains close to the reference policy (the second term of KL divergence):

$$\mathcal{J}_{\text{GRPO}}(f_{\boldsymbol{\theta}}) \triangleq \mathbb{E}_{(q,a)\sim\mathcal{D},\{o_i\}_{i=1}^{G}\sim f_{\text{old}}(\cdot|q)} \frac{1}{G} \sum_{i=1}^{G} \frac{1}{|o_i|} \sum_{t=1}^{|o_i|}$$

$$\left[ \min\left( \frac{f_{\boldsymbol{\theta}}(o_{i,t}|q, o_{i,<t})}{f_{\text{old}}(o_{i,t}|q, o_{i,<t})} \hat{A}_i, \text{clip}\left( \frac{f_{\boldsymbol{\theta}}(o_{i,t}|q, o_{i,<t})}{f_{\text{old}}(o_{i,t}|q, o_{i,<t})}, 1-\epsilon, 1+\epsilon \right) \hat{A}_i \right) - \beta \mathbb{D}_{\text{KL}}\left[ f_{\boldsymbol{\theta}}||f_{\text{ref}} \right] \right].$$

Here, the $\text{clip}(\cdot, 1-\epsilon, 1+\epsilon)$ ensures that updates do not deviate excessively from the old policy by bounding the policy ratio between $1-\epsilon$ and $1+\epsilon$. Besides, the KL divergence is estimated as:

$$\mathbb{D}_{\text{KL}}\left[ f_{\boldsymbol{\theta}}||f_{\text{ref}} \right] = \frac{f_{\text{ref}}(o_{i,t}|q, o_{i,<t})}{f_{\boldsymbol{\theta}}(o_{i,t}|q, o_{i,<t})} - \log \frac{f_{\text{ref}}(o_{i,t}|q, o_{i,<t})}{f_{\boldsymbol{\theta}}(o_{i,t}|q, o_{i,<t})} - 1.$$

Nonetheless, GRPO can be challenging to implement because it sometimes produces outputs with unintended token distributions or incoherent language patterns (Guo et al., 2025). It also demands careful reward function design to balance fairness constraints and meaningful group-based advantage estimation (Stiennon et al., 2020). The RL training process in GRPO can be unstable due to its reliance on group-based relative advantages, and it remains computationally expensive,

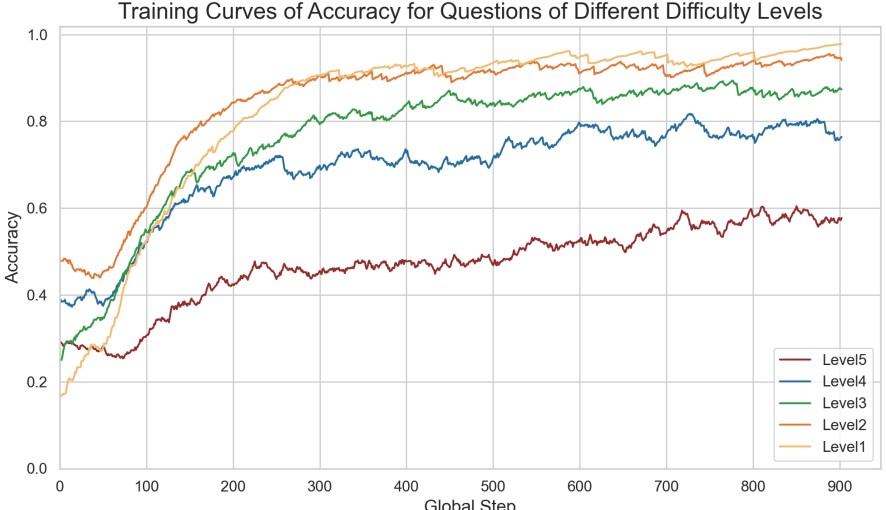

*Figure 4.* The GRPO training accuracy curves for questions of different difficulty levels on Qwen2.5-Math-1.5B.

especially for large-scale implementations (Schulman et al., 2017; Ouyang et al., 2022). Furthermore, while GRPO introduces optimizations to post-training, it does not consistently outperform simpler methods like SFT, particularly in small-scale training or with smaller models. This highlights the trade-offs between complexity, computational cost, interpretability, and practical effectiveness (Ouyang et al., 2022).

Given the limitations of base models (Li et al., 2023b; Zhou et al., 2024; 2025c; 2026), several RL algorithms have been developed primarily for alignment tasks. Therein, DPO (Rafailov et al., 2023), CPO (Xu et al., 2024), and their variants (Li et al., 2024; Guo et al., 2024; Munos et al., 2024; Hong et al., 2024; Xie et al., 2025a) rely on pairs of outputs labeled by human preference. In contrast, KTO (Ethayarajh et al., 2024) and BCO (Jung et al., 2025) require only a single binary label (like or dislike) for each output. Besides, the PRM (Uesato et al., 2022; Lightman et al., 2024) and Step-KTO (Lin et al., 2025a) offer step-by-step guidance by incorporating feedback at each reasoning step rather than focusing solely on the final outputs. Recently, the follow-up work of GRPO improves the optimization objective, *e.g.*, DAPO (Yu et al., 2025), Dr. GRPO (Liu et al., 2025a), REINFORCE++ (Hu et al., 2025), CPPO (Lin et al., 2025b), and GPG (Chu et al., 2026). Another line of research generalizes GRPO to broader applications such as multimodal reasoning (Zhou et al., 2025a; Huang et al., 2026; Chu et al., 2026; Liu et al., 2025b; Zhang et al., 2025), multi-agent reasoning (Xie et al., 2025c), logical reasoning (Xie et al., 2025b), self-supervised reasoning (Zhang et al., 2026), molecular optimization (Li et al., 2026), and agentic reasoning (Zhou et al., 2025b).

## B. Full Results and Analysis of the Training Dynamics

In this section, we provide the full results and analysis of the training dynamics in Sec. 3.

### B.1. Difficulty Estimation

Accurate estimation of question difficulty plays a crucial role in our proposed algorithm, CoDaPO, as it directly influences the computation of difficulty-adaptive weights in the optimization objective. However, relying solely on pre-existing difficulty annotations presents significant limitations. First, not all datasets contain ground-truth difficulty labels. Second, since the model's performance evolves during training, the perceived difficulty of a question may vary over time. Consequently, fixed difficulty labels may fail to reflect the dynamic nature of the model's learning process.

To explore a more adaptive and robust difficulty estimation approach, we conduct preliminary experiments on the MATH dataset, which includes human-annotated difficulty levels. Using GRPO on Qwen2.5-Math-1.5B, we analyze the accuracy trajectories for different difficulty levels throughout training. As shown in Fig. 4, model accuracy aligns well with the ground-truth difficulty labels: easier questions correspond to higher accuracy, and harder questions to lower accuracy.

| Confidence | Step 0 | | | Step 60 | | | Step 240 | | |
|---|---|---|---|---|---|---|---|---|---|
| | Level 1 | Level 3 | Level 5 | Level 1 | Level 3 | Level 5 | Level 1 | Level 3 | Level 5 |
| Minimum | 0.00 | 0.00 | 0.00 | 0.01 | 0.00 | 0.00 | 0.32 | 0.54 | 0.36 |
| Maximum | 0.99 | 0.99 | 0.99 | 0.99 | 0.98 | 0.99 | 0.98 | 0.97 | 0.98 |
| Mean | 0.78 | 0.75 | 0.69 | 0.82 | 0.81 | 0.76 | 0.86 | 0.87 | 0.86 |
| Std | 0.15 | 0.18 | 0.21 | 0.10 | 0.14 | 0.17 | 0.06 | 0.05 | 0.06 |
| Median | 0.83 | 0.81 | 0.75 | 0.84 | 0.84 | 0.81 | 0.87 | 0.88 | 0.87 |
| Kurtosis | 6.69 | 2.53 | 0.60 | 16.10 | 8.41 | 3.32 | 10.02 | 2.67 | 5.47 |

*Table 12.* Confidence statistics at different steps and difficulty levels.

| Gradient | Step 0 | | | Step 60 | | | Step 240 | | |
|---|---|---|---|---|---|---|---|---|---|
| | Level 1 | Level 3 | Level 5 | Level 1 | Level 3 | Level 5 | Level 1 | Level 3 | Level 5 |
| Minimum | 0.49 | 0.75 | 0.56 | 0.66 | 0.63 | 0.49 | 0.65 | 0.74 | 0.48 |
| Maximum | 1723.08 | 954.45 | 213.61 | 193.70 | 425.39 | 875.58 | 21.10 | 19.25 | 11.85 |
| Mean | 5.26 | 5.30 | 5.60 | 2.84 | 3.42 | 5.06 | 2.26 | 1.95 | 2.19 |
| Standard deviation | 42.90 | 26.05 | 11.51 | 6.78 | 12.75 | 25.66 | 1.14 | 0.89 | 0.90 |
| Median | 2.40 | 2.57 | 2.95 | 2.08 | 2.03 | 2.57 | 2.01 | 1.81 | 2.08 |
| Kurtosis | 1325.53 | 988.33 | 95.16 | 434.23 | 666.90 | 795.30 | 46.17 | 102.37 | 17.61 |

*Table 13.* Gradient statistics at different steps and difficulty levels.

Moreover, the accuracy gap between different difficulty levels increases and stabilizes as training progresses, indicating a consistent difficulty signal.

These observations motivate us to estimate question difficulty based on the model's own accuracy. This approach is naturally integrated into the online RL process without requiring any additional evaluation model, as multiple samples per question are generated during training. It generalizes well to datasets without difficulty annotations and provides a dynamic estimation mechanism that adapts to the model's evolving capabilities over time, overcoming the limitations of static difficulty labels.

### B.2. Quantitative Results of Training Dynamics

We show the quantitative results of Fig. 2 and Fig. 3 in Tab. 12 and Tab. 13, respectively. For each difficulty level, we fix a set of 100 validation questions and sample 20 trajectories per question from the model at each checkpoint. This yields exactly 2000 points per subplot. We repeat this procedure for the base model, for step 60, and for step 240 using the same questions and sampling protocol. Thus every subplot at every checkpoint is based on the full (and identical) validation subset, with 2000 points each. During GRPO training, model confidence progressively saturates toward 1, with both correct and incorrect responses clustering near high-confidence values, leading to miscalibration. Confidence differences across difficulty levels diminish over time, while tail behavior (kurtosis) reveals structural changes in the distribution. Gradients initially exhibit heavy-tailed bursts for easy questions, but training rapidly suppresses magnitudes and contracts distributions across all difficulty levels. By the end of training, gradient contraction becomes more uniform, though hard questions retain residual difficulty. Overall, GRPO induces saturation in confidence and global contraction in gradients, progressively reducing differences across difficulty levels.

**The full training dynamics.** In Figs. 5 and 6, we present the full confidence-advantage distribution and gradient distribution of different checkpoints during GRPO training on MATH dataset.

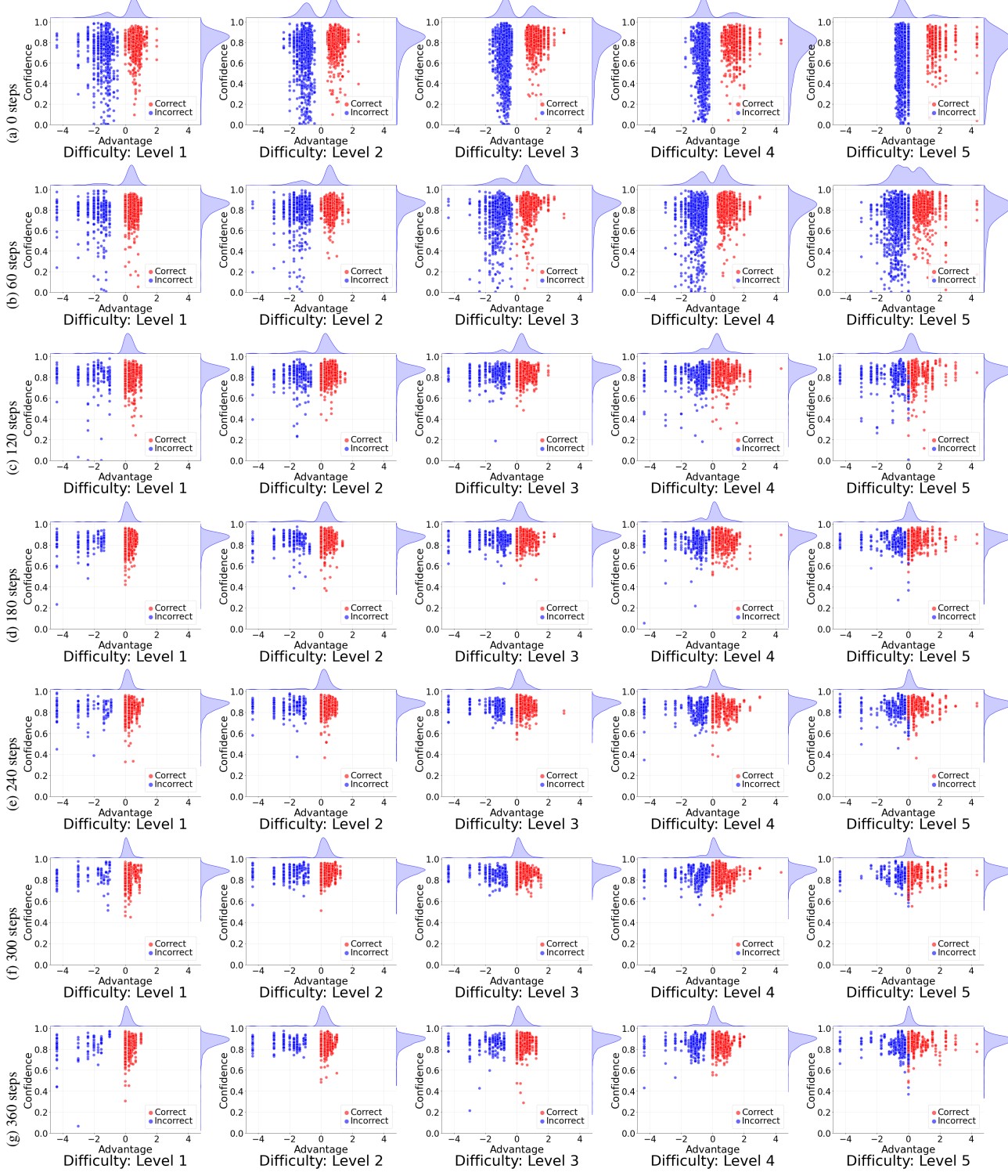

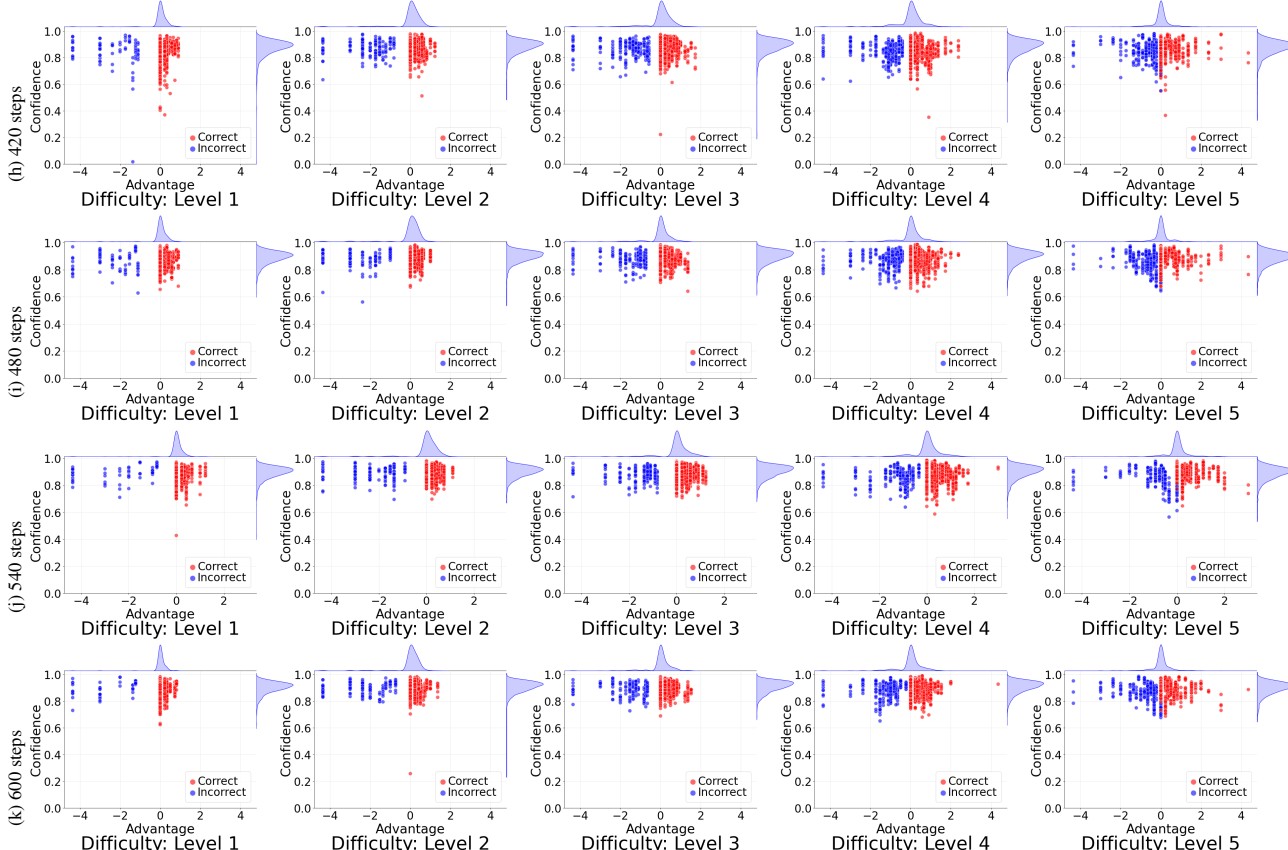

*Figure 5.* The confidence-advantage distribution of the Qwen2.5-Math-1.5B model, post-training on the MATH dataset with the GRPO algorithm. We present the distribution of model checkpoints captured at every 60 training steps. Columns 1 through 5 correspond to difficulty levels 1 through 5. Additionally, the marginal distributions (*i.e.*, the density distributions) of advantage and confidence across all samples are shown above and to the right of each confidence-advantage distribution.

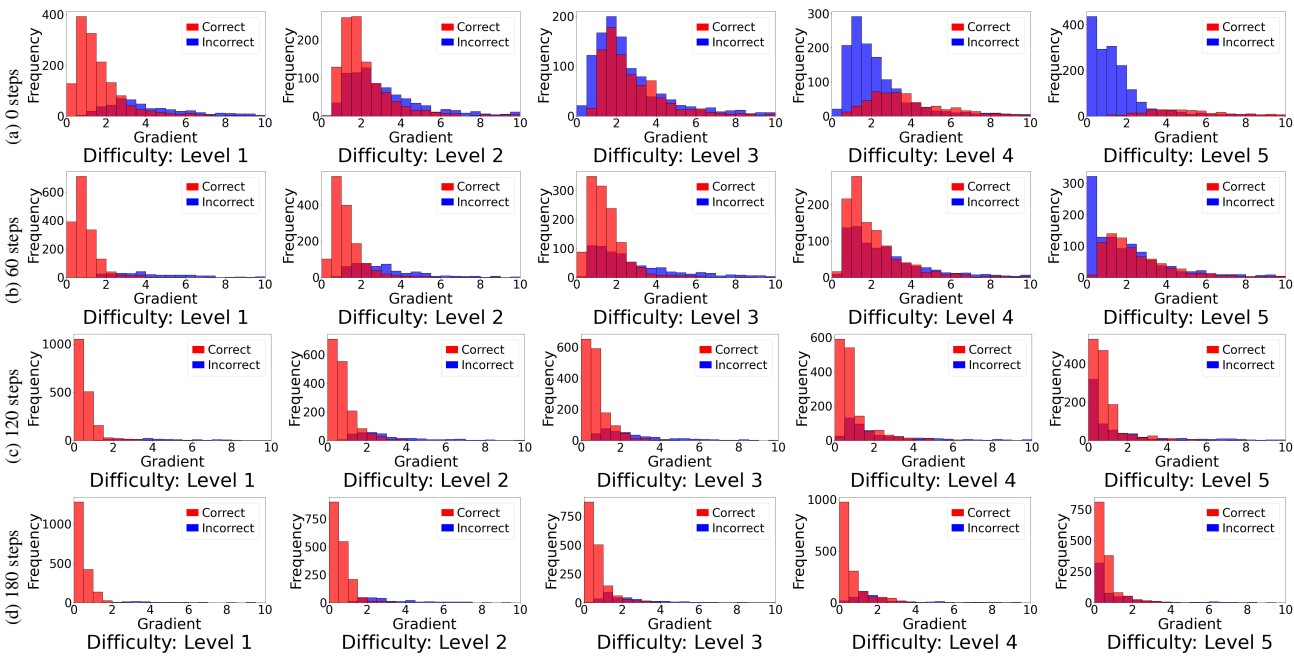

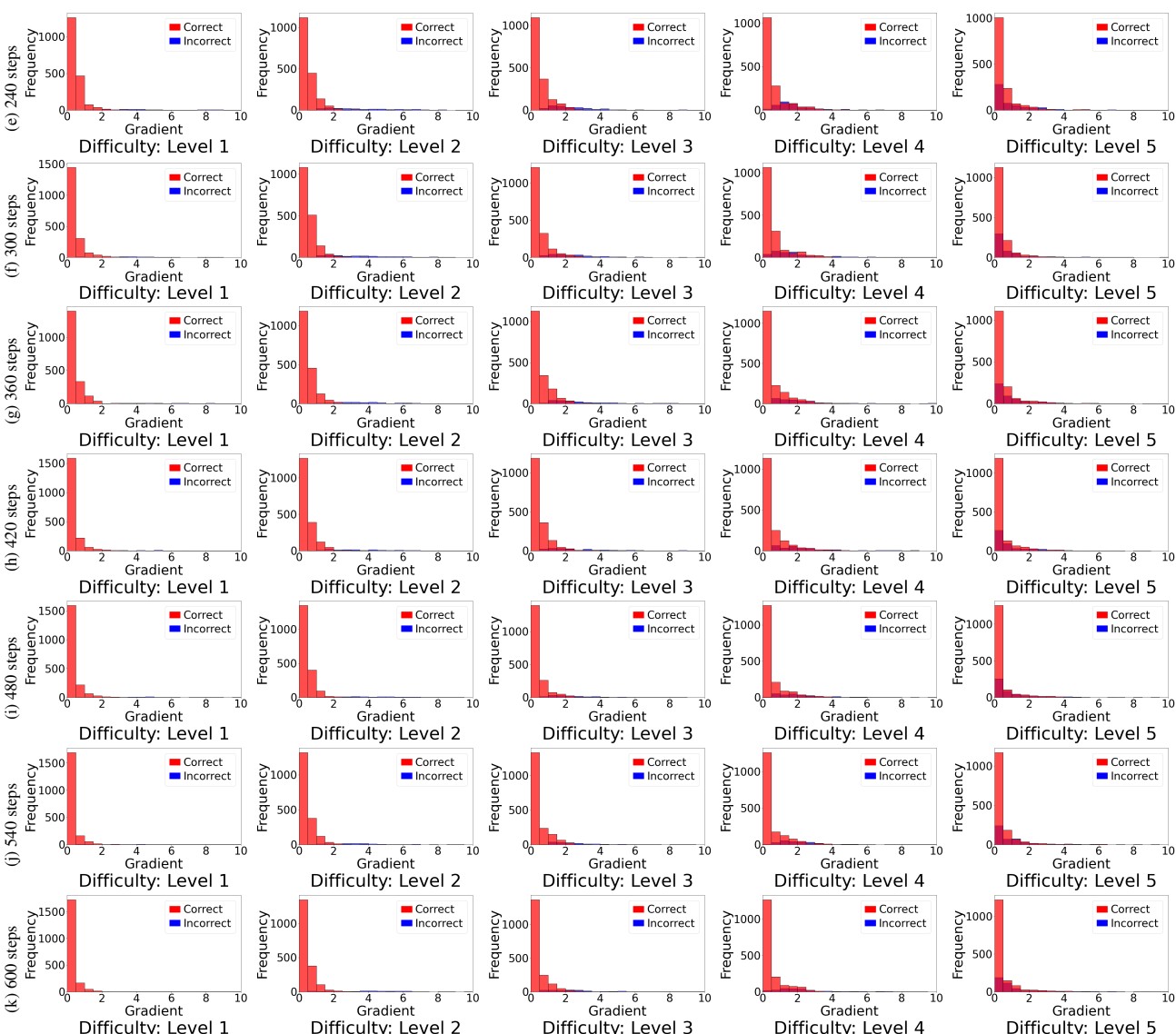

*Figure 6.* The gradient distribution of the Qwen2.5-Math-1.5B model, post-training on the MATH dataset with GRPO. We present the distribution of model checkpoints captured at every 60 training steps. Columns 1 through 5 correspond to difficulty levels 1 through 5. (consistent with Fig. 5).

## B.3. Why Probability Increases (and Entropy Collapses)

This subsection explains why GRPO tends to concentrate probability mass (lower entropy), often accompanied by overconfidence and miscalibration.

**Empirical observation.** Fig. 2 shows that during GRPO training, the model becomes increasingly *confident* on sampled trajectories: when mapped back to probability space (e.g., via $\exp(\texttt{Confidence})$, the geometric mean token probability), the distribution shifts toward near-certain predictions. Accuracy improves, but calibration degrades—the model also becomes more confident on *incorrect* trajectories. Compared to the base model (whose confidence in wrong answers is highly variable), the post-trained model concentrates probability mass for both correct and incorrect outputs, consistent with entropy collapse reported by Yu et al. (2025).

**Setup: per-token update direction.** A first-order ascent step on the token log-probability $\ell_{\boldsymbol{\theta}}(o_{i,t}) = \log f_{\boldsymbol{\theta}}(o_{i,t} \mid q, o_{i,<t})$ induced by GRPO can be written as

$$\Delta\ell_{\boldsymbol{\theta}}(o_{i,t}) \;\propto\; \underbrace{\mathbf{1}_{\text{unclipped}} \,\hat{A}_i\, \rho_{i,t}}_{\text{policy push}} \;-\; \underbrace{\beta\,(1 - u_{i,t})}_{\text{reference-KL pull}},$$

where $\rho_{i,t} = \exp(\ell_{\boldsymbol{\theta}}(o_{i,t}) - \ell_{\text{old}}(o_{i,t}))$ and

$$u_{i,t} \triangleq \frac{f_{\text{ref}}(o_{i,t} \mid q, o_{i,<t})}{f_{\boldsymbol{\theta}}(o_{i,t} \mid q, o_{i,<t})} = \exp\big(\ell_{\text{ref}}(o_{i,t}) - \ell_{\boldsymbol{\theta}}(o_{i,t})\big).$$

We next unpack how clipping creates an asymmetric drift, why the KL penalty mostly acts as a "floor" toward the reference, and how trajectory-level credit spreads these effects across tokens.

**1) Clipping creates an asymmetric drift toward higher probabilities.** For the PPO-style surrogate $\min(\rho_{i,t}\hat{A}_i,\ \text{clip}(\rho_{i,t})\hat{A}_i)$, the gradient is nonzero only when the *unclipped* branch is selected. This yields a key asymmetry:

- **Positive advantage ($\hat{A}_i > 0$).** The surrogate applies an *upward* update to $\ell_{\boldsymbol{\theta}}(o_{i,t})$ as long as $\rho_{i,t} \leq 1 + \epsilon$ (unclipped). Once $\rho_{i,t} > 1 + \epsilon$, the clipped branch is selected and the surrogate gradient becomes zero. Since $\rho_{i,t} = \exp(\ell_{\boldsymbol{\theta}} - \ell_{\text{old}})$, increasing $\ell_{\boldsymbol{\theta}}$ increases $\rho_{i,t}$, which in turn strengthens the push while it remains unclipped (a positive feedback loop) until it hits the cap $1 + \epsilon$.

- **Negative advantage ($\hat{A}_i < 0$).** The surrogate applies a *downward* update only while $\rho_{i,t} \geq 1 - \epsilon$ (unclipped). If $\rho_{i,t} < 1 - \epsilon$, the clipped branch is selected, the surrogate gradient becomes zero, so further decreases are not encouraged.

Therefore, increases in $\ell_{\boldsymbol{\theta}}$ can accumulate up to the upper ratio bound, whereas decreases are self-limiting once $\rho_{i,t}$ falls below $1 - \epsilon$. This induces a systematic upward bias in sampled-token probabilities and thus lowers entropy.

**2) The reference KL is bounded and mainly restores probabilities toward $f_{\text{ref}}$.** The KL term contributes

$$-\beta(1 - u_{i,t}) = -\beta\left(1 - \frac{f_{\text{ref}}(o_{i,t} \mid q, o_{i,<t})}{f_{\boldsymbol{\theta}}(o_{i,t} \mid q, o_{i,<t})}\right).$$

Its sign matches a restoring force toward the reference:

- If $f_{\boldsymbol{\theta}}(o_{i,t} \mid \cdot) > f_{\text{ref}}(o_{i,t} \mid \cdot)$, then $u_{i,t} < 1$ and the KL term is negative, pulling $\ell_{\boldsymbol{\theta}}$ downward.

- If $f_{\boldsymbol{\theta}}(o_{i,t} \mid \cdot) < f_{\text{ref}}(o_{i,t} \mid \cdot)$, then $u_{i,t} > 1$ and the KL term is positive, pushing $\ell_{\boldsymbol{\theta}}$ upward.

Two details matter in practice. First, the KL pull is *bounded* in magnitude (it approaches $+\infty$ only if $u_{i,t} \to \infty$, but for typical updates it is much smaller than the policy push unless $\beta$ is large). Second, when $\hat{A}_i < 0$ and $\rho_{i,t} < 1 - \epsilon$, the policy term is *inactive* while the KL term still pushes probabilities back toward the reference, preventing them from drifting too low. Net effect: KL often acts as a *floor* near $f_{\text{ref}}$, while positive-advantage updates repeatedly push probabilities upward until clipping.

**3) Trajectory-level credit assignment amplifies many tokens at once.** GRPO applies the same scalar advantage $\hat{A}_i$ to every token in a rollout. For high-reward trajectories, this means many tokens (including non-causal or stylistic ones) receive positive updates whenever they are unclipped. Consequently, probability mass concentrates broadly along the sampled trajectory, accelerating entropy reduction and sharpening confidence, even when the reward provides weak guidance about which tokens should change.

**Summary.** Entropy collapse in GRPO can be traced to (i) clipping-induced asymmetry that favors probability increases, (ii) a KL penalty that predominantly restores probabilities toward the reference (especially when the surrogate is clipped for negative advantages), and (iii) trajectory-level credit that spreads positive updates across many tokens. Together, these effects drive sampled-token probabilities toward near-deterministic values, including on trajectories that remain incorrect, yielding the observed overconfidence and miscalibration (Fig. 2).

### B.4. Why Advantage Contracts

**Empirical observation.** Fig. 2 shows that as GRPO improves accuracy, the *empirical* distribution of group-normalized advantages collapses toward 0: most samples end up with small (near-zero) advantages, while a shrinking fraction of failures carries increasingly large negative advantages. This appears as a growing spike near 0 and a thinner (rarer) negative tail.

**Mechanism: binary rewards plus group normalization.** For a fixed question $q$, each rollout $o_i$ receives a binary reward $r_i \in \{0, 1\}$. Let

$$\bar{r} \triangleq \frac{1}{G} \sum_{i=1}^{G} r_i \in [0, 1]$$

denote the group correctness rate. GRPO computes the (approximately) standardized advantage

$$\hat{A}_i = \frac{r_i - \bar{r}}{\sqrt{\bar{r}(1 - \bar{r}) + \delta}}, \qquad \delta > 0,$$

treating $\bar{r}$ and the denominator as a constant (stop-gradient). Since $r_i$ is binary, $\hat{A}_i$ takes only two values within a group. Ignoring $\delta$ for clarity, these values are

$$\hat{A}^{(+)}(\bar{r}) = \frac{1 - \bar{r}}{\sqrt{\bar{r}(1 - \bar{r})}} = \sqrt{\frac{1 - \bar{r}}{\bar{r}}}, \qquad \hat{A}^{(-)}(\bar{r}) = \frac{-\bar{r}}{\sqrt{\bar{r}(1 - \bar{r})}} = -\sqrt{\frac{\bar{r}}{1 - \bar{r}}}. \tag{7}$$

Eq. 7 directly explains the observed trend as $\bar{r}$ increases during training:

- **Correct samples shrink to zero.** If $\bar{r} \uparrow 1$, then $\hat{A}^{(+)}(\bar{r}) \downarrow 0$, so correct rollouts receive vanishing positive advantage.

- **Remaining errors become more negative.** If $\bar{r} \uparrow 1$, then $\hat{A}^{(-)}(\bar{r}) \to -\infty$, so the few remaining incorrect rollouts receive increasingly large negative advantage.

**Why pooling across groups produces a spike at** $0$. Within each group, standardization enforces (approximately) zero mean and unit variance. The key point is that the *pooled* distribution across many questions and training steps is a mixture whose *atom locations* and *weights* depend on $\bar{r}$. Conditioned on a given $\bar{r}$,

$$\hat{A} \mid \bar{r} \sim \bar{r} \delta_{\hat{A}^{(+)}(\bar{r})} + (1 - \bar{r}) \delta_{\hat{A}^{(-)}(\bar{r})}. \tag{8}$$

As training progresses, most groups shift toward $\bar{r} \approx 1$. Then (i) the dominant mass $\bar{r}$ concentrates near $\hat{A}^{(+)}(\bar{r}) \approx 0$, creating the spike at zero; and (ii) the negative atom moves left (more negative), but its weight $(1 - \bar{r})$ vanishes, making the tail rare. This resolves the apparent paradox: advantages are normalized *within each group*, yet their aggregated histogram still contracts because most groups become "almost always correct."

**Practical implication.** As $\hat{A}^{(+)}(\bar{r})$ shrinks, the effective policy-gradient signal on the majority of tokens diminishes, slowing further learning. Meanwhile, the few remaining errors carry large negative advantages but occur infrequently, yielding sparse and potentially unstable corrective updates, especially on hard questions.

## B.5. Why Training Converges Hierarchically

**Empirical observation.** Figs. 2 and 3 show a consistent "easy-to-hard" learning pattern. *Easy questions* quickly reach high confidence and high correctness; their advantages and gradient norms then decay rapidly toward zero. *Hard questions* start with low confidence and low correctness; training shifts them upward, but gradient norms still decay quickly and a substantial error mass remains for long.

**Mechanism: learning signal is gated by success frequency and capped by clipping.** For token $(i, t)$, define the scalar ascent weight (the coefficient of $\nabla_{\boldsymbol{\theta}} \ell_{\boldsymbol{\theta}}(o_{i,t})$)

$$w_{i,t} \triangleq \mathbf{1}_{\text{unclipped}} \, \hat{A}_i \, \rho_{i,t} \, - \, \beta \, (1 - u_{i,t}), \tag{9}$$

so that $w_{i,t} > 0$ increases $\ell_{\boldsymbol{\theta}}(o_{i,t})$ and $w_{i,t} < 0$ decreases it. Let $\bar{r}(q) \triangleq \frac{1}{G} \sum_{i=1}^{G} r_i$ be the group correctness rate for question $q$. Using the two-point form of GRPO advantages (Eq. 7), the per-question expected update can be decomposed as

$$\mathbb{E}[w_{i,t} \mid q] \approx \bar{r}(q) \, \mathbb{E}\Big[\mathbf{1}^{(+)}_{\text{unclipped}} \, \hat{A}^{(+)}(\bar{r}(q)) \, \rho_{i,t} \, \Big| \, r_i = 1, q\Big]$$
$$+ \, (1 - \bar{r}(q)) \, \mathbb{E}\Big[\mathbf{1}^{(-)}_{\text{unclipped}} \, \hat{A}^{(-)}(\bar{r}(q)) \, \rho_{i,t} \, \Big| \, r_i = 0, q\Big] \, - \, \beta \, \mathbb{E}[1 - u_{i,t} \mid q].$$

This expression highlights the core driver of hierarchical convergence: *useful positive updates require correct rollouts, whose frequency is controlled by $\bar{r}(q)$, and all updates are bounded by clipping.*

**Easy questions: frequent success $\Rightarrow$ rapid saturation and vanishing gradients.** For easy questions, $\bar{r}(q)$ becomes large early. Then correct rollouts dominate, and their advantage $\hat{A}^{(+)}(\bar{r}(q)) = \sqrt{\frac{1 - \bar{r}(q)}{\bar{r}(q)}}$ is small and decreases further as $\bar{r}(q) \uparrow 1$. Consequently:

- **Fast amplification.** Because correct rollouts are common, many tokens repeatedly receive positive updates until the ratio reaches the upper cap $\rho_{i,t} \approx 1 + \epsilon$.

- **Implicit annealing.** As $\bar{r}(q) \uparrow 1$, $\hat{A}^{(+)}(\bar{r}(q)) \downarrow 0$ (advantage contraction), so the policy-gradient magnitude on these questions shrinks even if the model remains correct.

- **Limited correction from rare failures.** Occasional incorrect rollouts have large negative $\hat{A}^{(-)}(\bar{r}(q))$, but they are down-weighted by $(1 - \bar{r}(q))$ and their surrogate gradient shuts off once $\rho_{i,t} < 1 - \epsilon$; afterward, the KL term mainly restores probabilities toward $f_{\text{ref}}$.

These effects explain why easy questions quickly move into a regime with near-saturated probabilities and rapidly vanishing gradient norms (Fig. 3).

**Hard questions: rare success $\Rightarrow$ slow discovery followed by capped reinforcement.** For hard questions, $\bar{r}(q)$ is small initially, so correct rollouts are rare. Within a group of $G$ samples, the probability of observing at least one correct rollout is

$$\mathbb{P}(\exists i : r_i = 1 \mid q) = 1 - (1 - \pi(q))^G \approx G \, \pi(q) \quad \text{for small } \pi(q),$$

where $\pi(q) \triangleq \mathbb{P}_{o \sim f_{\text{old}}(\cdot \mid q)}[r(o) = 1]$ is the per-sample success probability under the behavior policy. If no correct rollout is discovered, the learning signal is dominated by incorrect rollouts and the KL term; because negative updates are clipped once $\rho_{i,t} < 1 - \epsilon$, these forces tend to keep the policy near the behavior/reference rather than substantially increasing success. When a correct rollout *is* discovered, its advantage $\hat{A}^{(+)}(\bar{r}(q))$ can be large (since $\bar{r}(q)$ is small), producing a strong positive update—but only until clipping caps it at $\rho_{i,t} \approx 1 + \epsilon$. Thus progress on hard questions naturally takes two phases:

- **Discovery:** rare appearance of a correct trajectory (probability $\approx G \, \pi(q)$),

- **Amplification:** reinforcing the discovered trajectory up to the clipping cap, which gradually increases $\pi(q)$ over time.

This yields slow improvement and a persistent error mass, even as confidence gradually increases.

We note that the independence assumption is a simplification, while the exact within-group correlation is intractable. Modeling this correlation depends on output distribution, temperature, and prompt. We adopt the i.i.d. formulation to convey the core intuition: discovery probability grows with $G$ and $\pi(q)$. We also note that the i.i.d. assumption yields an optimistic estimate of discovery probability: the within-group correlation in practice means multiple rollouts tend to fail in similar ways, reducing effective sample diversity below $G$ independent draws. The true discovery probability is therefore lower than $1 - (1 - \pi(q))^G$, making the discovery bottleneck more challenging than Eq. 2 suggests. This confirms that allocating additional independent groups to high-value questions is a well-motivated strategy. In addition, Pass@K results empirically validate the discovery probability framework. Tab. 4 shows CoDaPO achieves higher Pass@K than GRPO at all K, validating the improved $\pi(q)$.

**Why the overall training looks hierarchical.**  Because easy questions generate dense correct samples early, they are reinforced quickly and then enter a vanishing-signal regime due to advantage contraction and clipping. Hard questions, in contrast, rely on sparse discovery events and capped amplification. As more easy questions saturate, a growing fraction of samples contribute little gradient (Fig. 3), leaving hard questions with limited effective updates. This mismatch in learning-signal density (easy: frequent and quickly annealed; hard: rare and capped) produces the observed hierarchical convergence under GRPO.

## C. Full Analysis of CoDaPO

This section provides a mechanism analysis of CoDaPO and explicitly links each component to the training dynamics diagnosed in Sec. 3 and Appendix B. Throughout, we emphasize how CoDaPO reallocates learning signal across *tokens* (micro-averaging), *questions* (value weighting), and *compute* (value-guided resampling), thereby addressing three phenomena observed for GRPO-style training: confidence inflation (entropy collapse), advantage contraction, and hierarchical (easy-to-hard) convergence.

**Preliminaries and conventions.**  Fix a mini-batch $\mathcal{B} = \{(q^{(j)}, a^{(j)})\}_{j=1}^B$ and, for each question $q^{(j)}$, a rollout group $\{o_i^{(j)}\}_{i=1}^G \sim f_{\text{old}}(\cdot \mid q^{(j)})$. For token position $t$ in rollout $o_i^{(j)}$, define

$$\ell_{\boldsymbol{\theta}}(o_{i,t}^{(j)}) \triangleq \log f_{\boldsymbol{\theta}}(o_{i,t}^{(j)} \mid q^{(j)}, o_{i,<t}^{(j)}), \qquad \rho_{i,t}^{(j)} \triangleq \frac{f_{\boldsymbol{\theta}}(o_{i,t}^{(j)} \mid q^{(j)}, o_{i,<t}^{(j)})}{f_{\text{old}}(o_{i,t}^{(j)} \mid q^{(j)}, o_{i,<t}^{(j)})}.$$

Each rollout receives a binary accuracy reward $r_i^{(j)} \in \{0, 1\}$, and we compute the group-normalized advantage

$$\hat{A}_i^{(j)} \triangleq \frac{r_i^{(j)} - \text{mean}(\{r_\ell^{(j)}\}_{\ell=1}^G)}{\text{std}(\{r_\ell^{(j)}\}_{\ell=1}^G)}.$$

As in Appendix B, let $\bar{r}^{(j)} \triangleq \frac{1}{G}\sum_{i=1}^G r_i^{(j)}$. Ignoring the numerical stabilizer for clarity, binary rewards imply the two-point values

$$\hat{A}^{(+)}(\bar{r}) = \sqrt{\frac{1 - \bar{r}}{\bar{r}}}, \qquad \hat{A}^{(-)}(\bar{r}) = -\sqrt{\frac{\bar{r}}{1 - \bar{r}}}.$$

We adopt the standard implementation convention used in our experiments: *stop gradient* through (i) $\hat{A}_i^{(j)}$ (including the group mean and standard deviation) and (ii) the CoDaWeighting value $v^{(j)}$ (including $c_q, d_q$). This isolates the effect of CoDaPO on the policy-gradient *weights*.

### C.1. CoDaLearning: Gradient Structure and Effective Token Weights

**Clipped coefficient and micro-averaging.**  Define the clipped coefficient

$$g(\rho, \hat{A}) \triangleq \min\left(\rho\hat{A}, \text{ clip}(\rho, 1 - \epsilon, 1 + \epsilon)\hat{A}\right). \tag{10}$$

The CoDaPO objective in Eq. 6 is a token micro-average of $v^{(j)}g(\rho_{i,t}^{(j)}, \hat{A}_i^{(j)})$ across all tokens in all rollouts in the batch.

**Subgradient form.** Because $\nabla_{\boldsymbol{\theta}}\rho_{i,t}^{(j)} = \rho_{i,t}^{(j)}\nabla_{\boldsymbol{\theta}}\ell_{\boldsymbol{\theta}}(o_{i,t}^{(j)})$, a (sub)gradient of Eq. 6 can be written as

$$\nabla_{\boldsymbol{\theta}}\mathcal{J}_{\text{CoDaPO}} = \sum_{j=1}^{B}\frac{1}{\sum_{i=1}^{G}|o_i^{(j)}|}\sum_{i=1}^{G}\sum_{t=1}^{|o_i^{(j)}|}w_{i,t}^{(j)}\,\nabla_{\boldsymbol{\theta}}\ell_{\boldsymbol{\theta}}(o_{i,t}^{(j)}), \tag{11}$$

with the *effective token weight*

$$w_{i,t}^{(j)} \triangleq v^{(j)}\cdot\mathbf{1}_{\text{unclipped}}(\rho_{i,t}^{(j)},\hat{A}_i^{(j)})\cdot\rho_{i,t}^{(j)}\cdot\hat{A}_i^{(j)}. \tag{12}$$

Here $\mathbf{1}_{\text{unclipped}} = 1$ when the minimum in Eq. 10 is attained by the *unclipped* branch, and 0 otherwise (subgradient 0 on the clipped region). Eq. 12 makes explicit that CoDaPO modifies GRPO through (i) micro-averaging (length-invariant aggregation) and (ii) a per-question multiplicative scaling $v^{(j)}$; CoDaSampling and the two-stage update then modify the *distribution* of questions and rollouts to which Eq. 12 is applied.

**Link to the dynamics in Sec. 3.** Appendix B shows that confidence inflation and hierarchical convergence are governed by repeated sign-consistent updates to $\ell_{\boldsymbol{\theta}}$, which are controlled by the magnitude and sign of $w_{i,t}$. Thus, analyzing CoDaPO reduces to analyzing how its components reshape the distribution of $(v^{(j)}, \bar{r}^{(j)}, \rho_{i,t}^{(j)})$, and therefore the distribution of effective token weights Eq. 12.

## C.2. CoDaWeighting: Question-level Allocation via Confidence and Difficulty

**Confidence/difficulty statistics.** For a question $q$ and its rollout group $\{o_i\}_{i=1}^{G}$, CoDaWeighting computes

$$c_q \triangleq \exp\left[\frac{1}{G}\sum_{i=1}^{G}\frac{1}{|o_i|}\sum_{t=1}^{|o_i|}\log f_{\boldsymbol{\theta}}(o_{i,t}\mid q, o_{i,<t})\right], \qquad d_q \triangleq 1 - \frac{1}{G}\sum_{i=1}^{G}r_i,$$

so $c_q \in (0,1]$ and $d_q \in [0,1]$. The statistic $c_q$ corresponds to the geometric mean token probability and directly tracks the confidence inflation phenomenon in Sec. 3, while $d_q$ is the empirical group error rate, consistent with the model-based difficulty used throughout Sec. 3.

**Value function and its regularity.** CoDaPO assigns

$$v_q \triangleq V(c_q, d_q) = V_c(c_q)\,V_d(d_q), \qquad V_c(x) = x, \quad V_d(x) = 1 - 4(x - \tfrac{1}{2})^2.$$

This choice satisfies $v_q \in [0,1]$ and is smooth on $[0,1]^2$; moreover $V_d$ is maximized at $d = \frac{1}{2}$ and vanishes at $d \in \{0,1\}$. Consequently, CoDaWeighting induces a bounded, "learnability"-oriented allocation: it prioritizes questions that are neither solved nor hopeless under the current behavior policy.

**Geometric mean as confidence.** The geometric mean naturally captures per-token commitment along a trajectory. It is also (1) already computed for GRPO's importance ratio (zero extra cost), (2) bounded in $(0,1]$ for multiplicative weighting, and (3) directly tied to the confidence inflation diagnosed in Sec. 3.

**Comparison with alternative confidence measures.** Output entropy requires the full distribution over all possible continuations; $c_q$ only needs the probability of actually generated tokens, directly distinguishing deliberate reasoning from random guessing. Answer consistency is a coarser, outcome-level signal that cannot separate committed but incorrect reasoning from guessing.

**Effect on expected update magnitude across questions.** Summing Eq. 11 over tokens and taking conditional expectation given $q$ yields

$$\mathbb{E}\left[\sum_{i,t}w_{i,t}^{(j)}\,\Bigg|\,q^{(j)}\right] = v^{(j)}\cdot\mathbb{E}\left[\sum_{i,t}\mathbf{1}_{\text{unclipped}}\,\rho_{i,t}^{(j)}\,\hat{A}_i^{(j)}\,\Bigg|\,q^{(j)}\right],$$

i.e., $v^{(j)}$ is an explicit *question-level multiplier* on the expected magnitude of policy updates for $q^{(j)}$.

**Connection to advantage contraction.** Appendix B shows that as $\bar{r} \uparrow 1$ (easy questions), $\hat{A}^{(+)}(\bar{r}) \downarrow 0$, so the learning signal on correct trajectories vanishes. CoDaWeighting reinforces this implicit annealing: when $d_q = 1 - \bar{r}$ is small, $V_d(d_q) \approx 0$, thereby further reducing compute allocated to already-solved questions. This mitigates the tendency of uniform training to keep spending updates on easy items even when their advantages have contracted.

**Connection to hierarchical convergence.** Hierarchical convergence arises because hard questions are discovery-limited: correct rollouts occur rarely and optimization is dominated by weak or clipped negative updates (Appendix B). For extremely hard questions, $d_q \approx 1$ and thus $V_d(d_q) \approx 0$, which intentionally deprioritizes regimes where discovery is negligible and gradients provide little progress. Instead, CoDaWeighting emphasizes intermediate difficulty ($d_q \approx \frac{1}{2}$), where correct trajectories appear often enough to provide a usable positive signal, but errors remain common enough that further improvement is possible. In this sense, CoDaWeighting operationalizes the "learnable band" implicit in the hierarchical convergence analysis.

**Connection to confidence inflation (entropy collapse).** Appendix B attributes confidence inflation to frequent positive updates on already-correct trajectories (especially for easy questions) and clipping asymmetry. Since CoDaWeighting suppresses $d_q \approx 0$ questions via $V_d(d_q) \approx 0$, it reduces the fraction of updates spent on saturated, already-solved items that chiefly contribute to further confidence inflation. We emphasize that CoDaPO does not eliminate the underlying asymmetry; rather, it reallocates compute away from the regime where the inflation is least informative.

### C.3. CoDaSampling: Value-guided Compute Reallocation and Discovery Amplification

**Induced sampling distribution.** Given values $\{v^{(j)}\}_{j=1}^B$ on a batch $\mathcal{B}$, let $\mathcal{I}_K(\mathcal{B})$ denote the indices of the top-$K$ items. CoDaSampling forms $\mathcal{S}$ by sampling with replacement from $\{(q^{(j)}, a^{(j)}) : j \in \mathcal{I}_K(\mathcal{B})\}$. Conditionally on $\mathcal{B}$, each draw selects item $j$ with probability

$$\mathbb{P}(J = j \mid \mathcal{B}) = \frac{1}{K} \mathbf{1}[j \in \mathcal{I}_K(\mathcal{B})].$$

When $K$ divides $B$, each top-$K$ item appears exactly $m = B/K$ times in $\mathcal{S}$.

**Effect on discovery probability.** Let $\pi(q) \triangleq \mathbb{P}_{o \sim f_{\text{old}}(\cdot|q)}[r(o) = 1]$ be the per-rollout success probability under the behavior policy. Appendix B shows that within one group of size $G$,

$$p_{\text{disc}}(q) \triangleq \mathbb{P}(\exists i : r_i = 1 \mid q) = 1 - (1 - \pi(q))^G.$$

If a question is repeated $m$ times in $\mathcal{S}$ and we resample fresh groups each time (as in Algorithm 1), the probability of observing at least one correct rollout in *any* of the $m$ groups is

$$1 - \left(1 - p_{\text{disc}}(q)\right)^m = 1 - \left((1 - \pi(q))^G\right)^m = 1 - (1 - \pi(q))^{Gm}. \tag{13}$$

Therefore, value-guided repetition increases the discovery probability as if the group size were effectively scaled from $G$ to $Gm$. This provides a formal bridge to hierarchical convergence: CoDaSampling increases the rate of "discovery events" that unlock positive-advantage updates on questions that are currently learnable (high $v_q$).

**Interaction with advantage contraction.** As training progresses, questions with $d_q \approx 0$ tend to have small $v_q$ and thus are unlikely to remain in the top-$K$ set. Hence CoDaSampling automatically shifts compute away from questions whose gradients have already annealed due to advantage contraction, reducing wasted rollouts and updates.

### C.4. Two-stage CoDaLearning: Coverage and Concentration

**Decomposition of one-step update.** Let $\nabla \mathcal{J}_\mathcal{B}$ and $\nabla \mathcal{J}_\mathcal{S}$ denote the stochastic gradients produced by applying `CoDaLearning` to $(\mathcal{B}, \mathcal{O}_\mathcal{B}, \mathcal{V}_\mathcal{B})$ and $(\mathcal{S}, \mathcal{O}_\mathcal{S}, \mathcal{V}_\mathcal{S})$, respectively. One training step follows the combined direction

$$\Delta \boldsymbol{\theta} \propto \nabla_{\boldsymbol{\theta}} \mathcal{J}_\mathcal{B} + \nabla_{\boldsymbol{\theta}} \mathcal{J}_\mathcal{S}.$$

The batch-wide term preserves coverage over $\mathcal{B}$ (reducing sensitivity to noise in $v$ and preventing selection collapse), while the focused term concentrates computation on high-value questions (increasing effective sample size and amplifying discovery as in Eq. 13).

**Why resampling fresh rollouts improves stability.** The value $v^{(j)}$ is estimated from $\mathcal{O}_\mathcal{B}$, whereas the focused update uses fresh rollouts $\mathcal{O}_\mathcal{S}$. This separation reduces coupling between selection noise and optimization noise: the focused gradient is not conditioned on the particular rollouts used to compute the values, which empirically improves stability and reduces overfitting to a fixed set of sampled trajectories.

## C.5. Micro-averaging: Length-invariant Credit Assignment

**Eliminating length-induced scaling.** In standard per-trajectory averaging, each rollout contributes a factor $1/|o_i|$, which implicitly down-weights longer outputs and can create incentives to shorten generations (Sec. 3). CoDaPO instead micro-averages over tokens by normalizing with $\sum_i |o_i|$ (within each question group), so each token has equal weight in Eq. 11. This preserves the per-token update directions (controlled by $w_{i,t}^{(j)}$) while removing output length as a confounder in the magnitude of the stochastic gradient.

## C.6. CoDaPO as a Data-Centric Compute Allocation Framework

CoDaPO is a data-centric compute allocation framework, not merely a reweighting scheme or an engineering recipe. Its design follows a diagnosis-to-method pipeline: empirical observation (Sec. 3.2) $\rightarrow$ mathematical analysis (Sec. 3.3) $\rightarrow$ targeted component design (Sec. 4).

**Beyond loss reweighting.** CoDaWeighting suppresses easy questions ($d_q \approx 0$, advantage contraction) and modulates updates via confidence against confidence inflation. Beyond this, CoDaSampling reallocates rollout compute by selecting high-value questions and generating *fresh rollouts*, increasing discovery probability from $1-(1-\pi(q))^G$ to $1-(1-\pi(q))^{Gm}$. This data-level intervention changes *which questions the model trains on*, directly addressing the discovery bottleneck for hard questions. CoDaLearning further introduces a two-stage update, enabling both broad coverage and focused refinement. KL removal and micro-averaging are auxiliary choices, consistent with prior work, and not claimed as core contributions.

**Confidence and difficulty as compute allocation signals.** The confidence signal $c_q$ and difficulty signal $d_q$ serve as relative indicators of trajectory informativeness, not calibrated probabilities. Vanilla GRPO yields uninformative updates on easy questions (vanishing advantages), hard questions (no correct rollouts), and overconfident wrong answers (confidence inflation). CoDaPO combines both signals via $v_q = c_q \cdot (1 - 4(d_q - 1/2)^2)$ to identify *which questions are currently most informative*: $d_q$ localizes the learnable band while $c_q$ distinguishes deliberate reasoning from random guessing within that band. Since both are computed from quantities already available in GRPO (log-probabilities for $c_q$, group rewards for $d_q$), they require zero additional inference cost.

**Functional form design.** The functional forms are derived from the diagnosed training dynamics. $V_d = 1 - 4(d - 1/2)^2$ suppresses both extremes ($d \approx 0$: vanishing advantage signal; $d \approx 1$: no correct discovery) and peaks at mid-difficulty, the simplest smooth function satisfying these constraints. $V_c = c$ upweights committed trajectories worth refining. The resulting value $v_q$ is fully specified from a single rollout group and requires no normalization: the same scalar drives both CoDaWeighting (multiplicative factor) and CoDaSampling (Top-K ranking). Algorithm 1 provides complete pseudocode.

## C.7. Summary

**Mechanistic correspondence to Sec. 3.** CoDaPO modifies GRPO through a minimal set of operations with explicit mathematical effects:

- **Confidence inflation (entropy collapse).** CoDaPO does not alter the clipping asymmetry analyzed in Appendix B, but it reduces uninformative updates that exacerbate saturation by down-weighting and resampling away from $d_q \approx 0$ questions.

- **Advantage contraction.** Since $\hat{A}^{(+)}(\bar{r}) \downarrow 0$ as $\bar{r} \uparrow 1$, CoDaPO further deallocates compute from such groups via $V_d(d_q) \approx 0$ and top-$K$ selection, making contraction an explicit annealing mechanism rather than a source of wasted updates.

- **Hierarchical convergence.** For questions in the learnable regime, value-guided repetition increases discovery probability from $1 - (1 - \pi)^G$ to $1 - (1 - \pi)^{Gm}$ by Eq. 13, accelerating the discovery–amplification cycle described in Appendix B.

We summarize the objective components in different RL algorithms for comparison in Tab. 14. Overall, CoDaPO implements a bounded, model-adaptive compute allocation strategy while retaining the stability properties of clipped policy optimization.

# D. Experiment Details

**Benchmarks.** We evaluate our proposed method and other baselines on the following diverse benchmarks.

| Normalization | Advantage | Regularization | Note |
|---|---|---|---|
| **GRPO (Shao et al., 2024)** | | | |
| $\mathbb{E}_{(q,a)\sim\mathcal{D},\{o_i\}_{i=1}^{G}\sim f_{\text{old}}(\cdot|q)}$ $\frac{1}{G}\sum_{i=1}^{G}\frac{1}{|o_i|}\sum_{t=1}^{|o_i|}\Big[$ | $\min\Big(\frac{f_{\boldsymbol{\theta}}(o_{i,t}|q,o_{i,<t})}{f_{\text{old}}(o_{i,t}|q,o_{i,<t})}\hat{A}_i,$ $\text{clip}(\frac{f_{\boldsymbol{\theta}}(o_{i,t}|q,o_{i,<t})}{f_{\text{old}}(o_{i,t}|q,o_{i,<t})},1-\epsilon,1+\epsilon)\hat{A}_i\Big)$ | $-\beta\Big(\frac{f_{\text{ref}}(o_{i,t}|q,o_{i,<t})}{f_{\boldsymbol{\theta}}(o_{i,t}|q,o_{i,<t})}$ $-\log\frac{f_{\text{ref}}(o_{i,t}|q,o_{i,<t})}{f_{\boldsymbol{\theta}}(o_{i,t}|q,o_{i,<t})}-1\Big)\Big].$ | $\hat{A}_i=\frac{r_i-\text{mean}(\{r_i\}_{i=1}^{G})}{\text{std}(\{r_i\}_{i=1}^{G})}.$ |
| **Dr. GRPO (Liu et al., 2025a)** | | | |
| $\mathbb{E}_{(q,a)\sim\mathcal{D},\{o_i\}_{i=1}^{G}\sim f_{\text{old}}(\cdot|q)}$ $\frac{1}{G\cdot c}\sum_{i=1}^{G}\sum_{t=1}^{|o_i|}\Big[$ | $\min\Big(\frac{f_{\boldsymbol{\theta}}(o_{i,t}|q,o_{i,<t})}{f_{\text{old}}(o_{i,t}|q,o_{i,<t})}\hat{A}_i,$ $\text{clip}(\frac{f_{\boldsymbol{\theta}}(o_{i,t}|q,o_{i,<t})}{f_{\text{old}}(o_{i,t}|q,o_{i,<t})},1-\epsilon,1+\epsilon)\hat{A}_i\Big)$ | $-\beta\Big(\frac{f_{\text{ref}}(o_{i,t}|q,o_{i,<t})}{f_{\boldsymbol{\theta}}(o_{i,t}|q,o_{i,<t})}$ $-\log\frac{f_{\text{ref}}(o_{i,t}|q,o_{i,<t})}{f_{\boldsymbol{\theta}}(o_{i,t}|q,o_{i,<t})}-1\Big)\Big].$ | $\hat{A}_i=r_i-\text{mean}(\{r_i\}_{i=1}^{G}).$ |
| **DAPO (Yu et al., 2025)** | | | |
| $\mathbb{E}_{(q,a)\sim\mathcal{D},\{o_i\}_{i=1}^{G}\sim f_{\text{old}}(\cdot|q)}$ $\frac{1}{\sum_{i=1}^{G}|o_i|}\sum_{i=1}^{G}\sum_{t=1}^{|o_i|}\Big[$ | $\min\Big(\frac{f_{\boldsymbol{\theta}}(o_{i,t}|q,o_{i,<t})}{f_{\text{old}}(o_{i,t}|q,o_{i,<t})}\hat{A}_i,$ $\text{clip}(\frac{f_{\boldsymbol{\theta}}(o_{i,t}|q,o_{i,<t})}{f_{\text{old}}(o_{i,t}|q,o_{i,<t})},1-\epsilon_{\text{low}},1+\epsilon_{\text{high}})\hat{A}_i\Big].$ | **None** | $\hat{A}_i=\frac{r_i-\text{mean}(\{r_i\}_{i=1}^{G})}{\text{std}(\{r_i\}_{i=1}^{G})}.$ |
| **GPG (Chu et al., 2026)** | | | |
| $\mathbb{E}_{(q,a)\sim\mathcal{D},\{o_i\}_{i=1}^{G}\sim f_{\text{old}}(\cdot|q)}$ $\frac{1}{\sum_{i=1}^{G}|o_i|}\sum_{i=1}^{G}\sum_{t=1}^{|o_i|}\Big[$ | $\log f_{\theta}(o_{i,t}|q,o_{i,<t})\hat{A}_i\Big].$ | **None** | $\hat{A}_i=\alpha\cdot\Big(r_i-\text{mean}(\{r_i\}_{i=1}^{G})\Big).$ |
| **CoDaPO (ours)** | | | |
| $\mathbb{E}_{(q,a)\sim\mathcal{D},\{o_i\}_{i=1}^{G}\sim f_{\text{old}}(\cdot|q)}$ $\frac{1}{\sum_{i=1}^{G}|o_i^{(j)}|}\sum_{i=1}^{G}\sum_{t=1}^{|o_i^{(j)}|}\Big[$ | $\min\Big(\frac{f_{\boldsymbol{\theta}}(o_{i,t}|q,o_{i,<t})}{f_{\text{old}}(o_{i,t}|q,o_{i,<t})}\hat{A}_i,$ $\text{clip}(\frac{f_{\boldsymbol{\theta}}(o_{i,t}|q,o_{i,<t})}{f_{\text{old}}(o_{i,t}|q,o_{i,<t})},1-\epsilon,1+\epsilon)\hat{A}_i\Big)v_q\Big].$ | **None** | $\hat{A}_i=\frac{r_i-\text{mean}(\{r_i\}_{i=1}^{G})}{\text{std}(\{r_i\}_{i=1}^{G})},$ $v_q=V(c_q,d_q)=c_q\Big(1-4(d_q-1/2)^2\Big),$ $c_q\triangleq\exp\Big[\frac{1}{G}\sum_{i=1}^{G}\frac{1}{|o_i|}\sum_{t=1}^{|o_i|}\log f_{\theta}(o_{i,t}\mid q,o_{i,<t})\Big],$ $d_q\triangleq 1-\frac{1}{G}\sum_{i=1}^{G}r_i.$ |

*Table 14.* Comparing the objective components in different RL algorithms for reasoning. Note that the **highlighted** contents indicate the differences with the original implementation of the GRPO algorithm. Specifically, (1) the advantage term evaluates the quality of model-generated outputs, (2) the regularization term measures the divergence between the policy model and a frozen reference model, and (3) the normalization term scales the final optimization signal across multiple problems and outputs.

1. **MATH 500 (Hendrycks et al., 2021b).** A curated set of 500 challenging problems from the MATH dataset, focusing on high school-level mathematics across algebra, geometry, number theory, and combinatorics.

2. **AIME 2024** [3]**.** A benchmark based on the 2024 American Invitational Mathematics Examination, testing advanced problem-solving skills with 15 short-answer math problems designed for top high school students.

3. **AIME 2025** [4]**.** The 2025 version of the AIME benchmark is similarly structured, providing a fresh set of high-difficulty pre-Olympiad level math problems.

4. **AMC 2023** [5]**.** Based on the 2023 American Mathematics Competitions (AMC 10/12), this benchmark assesses middle-to-advanced high school math across a range of topics in a multiple-choice format.

5. **OlympiadBench (He et al., 2024).** An olympiad-level scientific reasoning benchmark of bilingual, multimodal math and physics problems. Following common practice, we use its text-only English mathematics subset.

6. **Minerva (Lewkowycz et al., 2022).** A benchmark and model suite by Google that tackles math and science questions (from grade school to graduate level) using CoT reasoning and LLMs.

7. **GSM8K (Cobbe et al., 2021).** A dataset of 8,500 grade-school level math word problems designed to test models' ability to perform multi-step numerical reasoning in natural language.

8. **MMLU (Hendrycks et al., 2021a).** A large-scale multitask benchmark covering 57 academic subjects, ranging from STEM and humanities to social sciences, designed to evaluate broad knowledge understanding and reasoning abilities of language models.

9. **GPQA (Rein et al., 2024).** A benchmark of graduate-level, Google-proof question answering problems in physics, chemistry, and biology, specifically designed to assess expert-level scientific reasoning beyond simple retrieval.

---

[3] https://huggingface.co/datasets/HuggingFaceH4/aime_2024
[4] https://huggingface.co/datasets/opencompass/AIME2025
[5] https://huggingface.co/datasets/math-ai/amc23

10. **HumanEval (Chen et al., 2021).** A widely used code generation benchmark consisting of handwritten Python programming tasks, where models are evaluated based on functional correctness against hidden unit tests.

11. **TACO (Li et al., 2023a).** A benchmark for code generation and competitive programming that includes diverse programming tasks with varying difficulty levels, emphasizing both algorithmic reasoning and execution correctness.

12. **LiveCodeBench (Jain et al., 2025).** A continuously updated benchmark for evaluating code generation models on recent real-world competitive programming problems, focusing on robustness against benchmark contamination and temporal generalization.

**Experiment framework.** In this work, we utilize verl as the training backbone, specifically version 0.3.1. Building upon this implementation, we develop several baseline methods evaluated in this work, including DAPO, Dr. GRPO, GPG, and our proposed approach, CoDaPO. The evaluation adapts Qwen2.5-Math's evaluation codebase[6], ensuring consistent and reliable measurement across all experiments.

**General hyperparameters.** To ensure fair comparisons, we train all algorithms using the same set of hyperparameters.

- **Sampling setting.** For each question, we sample 8 responses to form a response group. The sampling temperature is set to 1.0, and we use a top-p value of 1.0 to consider the full token distribution.

- **Learning rate.** We use a learning rate of $1.0e - 6$ and apply no warm-up over the global training steps.

- **Batch size.** We set the batch size to 16 to evenly distribute the generated responses across training devices.

- **Randomness control.** To ensure reproducibility, we set the random seed to 42 and enable full determinism.

**Prompt template.** We use the prompt template shown in Fig. 7 for both training and evaluation. Furthermore, we apply chat template shown in Fig. 8 when processing the raw data.

---

**Prompt Template**

```
        Please reason step by step, and put your final answer within \boxed{}.
```

*Figure 7.* Prompt template used during training and evaluation.

---

**Chat Template**

```
<|im_start|>system
Please reason step by step, and put your final answer within \boxed{}.
<|im_end|>
<|im_start|>user
Find the sum of all integer bases $b>9$ for which $17_b$ is a divisor of $97_b$.
<|im_end|>
<|im_start|>assistant
```

*Figure 8.* Example of a prompt used after applying the chat template.

**Reward setting.** We observe that although using a format reward can improve the readability of LLM outputs to some extent, it may lead to reward hacking in the later stages of training, potentially resulting in irreversible training collapse. This observation is consistent with the findings of Zeng et al. (2025). Therefore, in all experiments conducted in this work, we rely solely on the accuracy reward, defined as follows:

$$r_i(y_i, \hat{y}) = \begin{cases} 1, & \text{is\_equivalent}(y_i, \hat{y}) \\ 0, & \text{otherwise} \end{cases}.$$

Here, $y_i$ denotes the answer produced for the $i$-th output and $\hat{y}$ represents the corresponding ground truth. To accurately determine whether $y_i$ and $\hat{y}$ are equivalent, all LaTeX expressions are parsed and compared for mathematical equivalence.

---

[6]https://github.com/QwenLM/Qwen2.5-Math/tree/main/evaluation

**Scaling to larger models.** We further evaluate CoDaPO on the larger Qwen2.5-14B-Instruct (Yang et al., 2024a). As shown in Tab. 15, while GRPO provides only marginal improvement over the base model on average ($45.33\% \rightarrow 45.61\%$), CoDaPO substantially boosts performance to $47.32\%$. In particular, CoDaPO achieves significant gains on challenging reasoning benchmarks, including +3.63% on AIME24, +1.65% on AIME25, and +3.50% on OlympiadBench over GRPO. These results suggest that the benefits of dynamics-aware compute allocation remain effective at larger model scales and become increasingly important for difficult reasoning tasks.

| Algorithm | MATH 500 | AIME 2024 | AIME 2025 | AMC 2023 | Olympiad Bench | Minerva | GSM8K | Avg |
|---|---|---|---|---|---|---|---|---|
| Base | 74.23 | 12.64 | 13.62 | 58.39 | 38.49 | 27.52 | 92.40 | 45.33 |
| GRPO | **75.91** | 10.57 | 15.55 | 59.73 | 38.13 | 26.13 | 93.27 | 45.61 |
| **CoDaPO** | 75.88 | **14.20** | **17.20** | **60.18** | **41.63** | **28.07** | **94.05** | **47.32** |

*Table 15.* Results on Qwen2.5-14B-Instruct.

**Compared with curriculum learning** CoDaPO is a dynamics-driven compute allocation framework which differs from curriculum learning in three key aspects:

1. **Non-monotonic weighting.** Curriculum learning typically follows a monotonic ordering (e.g., easy to hard). CoDaPO's U-shaped $V_d$ simultaneously suppresses both extremes ($d \approx 0$ and $d \approx 1$) and peaks at mid-difficulty, a non-monotonic design directly derived from GRPO's training dynamics.

2. **Joint confidence-difficulty signal.** CoDaPO uses $v_q = V_c(c_q) \cdot V_d(d_q)$, a joint signal from training dynamics analysis, across all three components (weighting, sampling, learning). This goes beyond difficulty-based data selection by incorporating the model's trajectory-level commitment.

3. **Beyond data ordering.** Curriculum learning selects or reorders training data. CoDaPO additionally performs gradient reweighting (CoDaWeighting) and two-stage policy updates (CoDaLearning), which cannot be replicated by any data scheduling strategy alone.

As shown in Tab. 10, neither single-level training nor linear curriculum matches CoDaPO, confirming that the gains stem from CoDaPO's design.

**Training tokens.** CoDaPO and GRPO are compared under the same total training step budget with identical per-step gradient compute in Section 5. The total training tokens are 105M for GRPO and 103M for CoDaPO, confirming comparable compute. The difference lies in dataset traversal: CoDaPO revisits high-value questions via resampling rather than always drawing new data.

## E. Case Studies

**In-domain case.** We analyze the performance gap between CoDaPO and GRPO through a symbolic reasoning task that requires algebraic manipulation, base conversion, and number-theoretic reasoning in Fig. 9. While both methods adopt similar initial steps, only CoDaPO successfully arrives at the correct and complete solution. This discrepancy reveals a broader insight: CoDaPO demonstrates a stronger capacity for maintaining symbolic consistency and handling algebraic constraints, whereas GRPO is more susceptible to local errors and brittle logic execution.

A key distinction lies in how the two models approach intermediate decision points. CoDaPO tends to preserve the symbolic structure of the problem throughout the reasoning process, producing interpretable and logically coherent derivations. In contrast, GRPO is more prone to heuristic or trial-based reasoning patterns, which may yield superficially plausible but ultimately incorrect results—especially in cases requiring discrete enumeration or careful constraint satisfaction.

This case exemplifies a common challenge in reinforcement learning for language models: small reasoning errors in early steps often cascade into incorrect final answers, and methods lacking robust symbolic understanding struggle to recover. CoDaPO mitigates this through more structured reasoning and better alignment between confidence and difficulty signals, which enhances its robustness in solving multi-step, discrete, and mathematically grounded problems.

These findings suggest that CoDaPO is not only effective in improving accuracy but also in enhancing reasoning fidelity and interpretability, especially in domains like mathematics, programming, and logic that require precise, symbolic manipulation.

**Out-of-domain case.** In tasks that require translating natural language descriptions into symbolic formulations grounded in domain knowledge—such as those involving physical systems or scientific laws (Fig. 10)—we observe a consistent performance gap between the two methods. CoDaPO demonstrates stronger semantic alignment between the problem description and the constructed solution model. Its responses tend to follow a coherent chain of physical and mathematical reasoning, with clear transitions from qualitative understanding to quantitative execution.

In contrast, GRPO is more prone to semantic drift: it often introduces irrelevant concepts, misapplies formulas, or incorrectly anchors symbolic steps to the problem's actual objective. This can lead to answers that are numerically plausible but logically invalid, due to disconnects between intermediate computations and the final question. Such behavior suggests difficulties in maintaining global consistency across multi-step derivations—particularly when domain-specific reasoning is required.

More generally, these observations highlight the importance of model capabilities in selective abstraction and structured model-building. An effective LLM solver must not only recall relevant formulas or operations, but also identify the appropriate scope of reasoning, exclude irrelevant detail, and maintain logical consistency through multi-step symbolic transformations. CoDaPO exhibits greater stability along these dimensions, making it more reliable on tasks that blend symbolic reasoning with physical interpretation.

**AIME25 Question**

Find the sum of all integer bases $b > 9$ for which $17_b$ is a divisor of $97_b$.

Ground Truth: $\boxed{70}$

**CoDaPO Response**

To solve this problem, we need to convert the numbers $17_b$ and $97_b$ from base $b$ to base 10 and then find the values of $b$ for which $17_b$ divides $97_b$.
1. Convert $17_b$ to base 10:

$$17_b = 1 \cdot b + 7 = b + 7.$$

2. Convert $97_b$ to base 10:

$$97_b = 9 \cdot b + 7 = 9b + 7.$$

We need $17_b$ to be a divisor of $97_b$, so:

$$9b + 7 \equiv 0 \pmod{b + 7}.$$

This means:
$$9b + 7 = k(b + 7)$$

for some integer $k$. Simplifying this, we get:

$$9b + 7 = kb + 7k$$
$$9b - kb = 7k - 7$$
$$b(9 - k) = 7(k - 1)$$

So:
$$b = \frac{7(k - 1)}{9 - k}.$$

Since $b > 9$, $k$ must be chosen such that $b$ is an integer greater than 9.
Let's find the valid values of $k$ and corresponding $b$.
We check the values of $k$ for which the right-hand side is an integer and $b > 9$:

- For $k = 2$: $b = \frac{7(2-1)}{9-2} = \frac{7}{7} = 1$, not greater than 9.

- For $k = 3$: $b = \frac{7(3-1)}{9-3} = \frac{14}{6} = \frac{7}{3}$, not an integer.

- For $k = 4$: $b = \frac{7(4-1)}{9-4} = \frac{21}{5}$, not an integer.

- For $k = 5$: $b = \frac{7(5-1)}{9-5} = \frac{28}{4} = 7$, not greater than 9.

- For $k = 6$: $b = \frac{7(6-1)}{9-6} = \frac{35}{3}$, not an integer.

- For $k = 7$: $b = \frac{7(7-1)}{9-7} = \frac{42}{2} = 21$, valid.

- For $k = 8$: $b = \frac{7(8-1)}{9-8} = \frac{49}{1} = 49$, valid.

- For $k = 9$: $b = \frac{7(9-1)}{9-9}$ is undefined.

- For $k = 10$: $b = \frac{7(10-1)}{9-10} = \frac{63}{-1} = -63$, not greater than 9.

So the valid values for $b$ are 21 and 49. The sum of these values is:

$$21 + 49 = 70.$$

Thus, the final answer is:
$\boxed{70}$

**GRPO Response**

To solve the problem, we need to translate the numbers $17_b$ and $97_b$ from base $b$ to base 10.
The number $17_b$ in base 10 is:

$$1 \cdot b + 7 = b + 7$$

The number $97_b$ in base 10 is:

$$9 \cdot b + 7 = 9b + 7$$

We need to find the values of $b$ such that $17_b$ (which is $b + 7$) is a divisor of $97_b$ (which is $9b + 7$). This means that:

$$b + 7 \mid 9b + 7$$

or equivalently:

$$9b + 7 \equiv 0 \pmod{b + 7}$$

We can express this congruence as an equation:

$$9b + 7 = k(b + 7)$$

for some integer $k$. Rearranging the equation:

$$9b + 7 = kb + 7k$$
$$9b - kb = 7k - 7$$
$$b(9 - k) = 7(k - 1)$$
$$b = \frac{7(k - 1)}{9 - k}$$

For $b$ to be an integer greater than 9, the right-hand side must be an integer. This means $7(k - 1)$ must be divisible by $9 - k$. We can iterate through possible integer values of $k$ to find valid solutions for $b$:

```
# Python code to find valid bases
valid_bases = []
for k in range(10): # Try k from 0 to 9
    if (9 - k) != 0:
        b = 7 * (k - 1) / (9 - k)
        if b.is_integer() and b > 9:
            valid_bases.append(int(b))

sum_of_bases = sum(valid_bases)
print((valid_bases, sum_of_bases))
```

Running this code yields:

$$\text{valid bases: } [16, 21] \quad \Rightarrow \quad \text{sum} = 16 + 21 = \boxed{37}$$

Therefore, the final answer is:

$\boxed{37}$

*Figure 9.* Case study of CoDaPO and GRPO Responses to an AIME25 Question.

## Minerva Question

An eclipsing binary consists of two stars of different radii and effective temperatures. Star 1 has radius $R_1$ and $T_1$, and Star 2 has $R_2 = 0.5R_1$ and $T_2 = 2T_1$. Find the change in bolometric magnitude of the binary, $\Delta m_{\text{bol}}$, when the smaller star is behind the larger star. (Consider only bolometric magnitudes so you don't have to worry about color differences.)

Ground Truth: $\boxed{1.75}$

### CoDaPO Response

To solve this problem, we need to calculate the change in bolometric magnitude of the binary system when the smaller star (Star 2) is behind the larger star (Star 1). The bolometric magnitude of a star is given by:

$$m_{\text{bol}} = -2.5 \log_{10}(L_{\text{bol}})$$

where $L_{\text{bol}}$ is the bolometric luminosity of the star. The bolometric luminosity of a star is proportional to its surface area and its effective temperature:

$$L_{\text{bol}} \propto R^2 T^4$$

So, the bolometric luminosities of Star 1 and Star 2 are:

$$L_{\text{bol,1}} \propto R_1^2 T_1^4$$

$$L_{\text{bol,2}} \propto R_2^2 T_2^4 = (0.5R_1)^2 (2T_1)^4$$
$$= 0.25R_1^2 \cdot 16T_1^4 = 4R_1^2 T_1^4$$

The total bolometric luminosity of the binary system when both stars are visible is:

$$L_{\text{bol,total}} = L_{\text{bol,1}} + L_{\text{bol,2}} = kR_1^2 T_1^4 + k \cdot 4R_1^2 T_1^4$$
$$= 5kR_1^2 T_1^4$$

When Star 2 is behind Star 1, the total bolometric luminosity is just the luminosity of Star 1:

$$L_{\text{bol,total}} = L_{\text{bol,1}} = kR_1^2 T_1^4$$

The change in bolometric magnitude is given by:

$$\Delta m_{\text{bol}} = m_{\text{bol,after}} - m_{\text{bol,before}}$$
$$= -2.5 \log_{10}\left(\frac{L_{\text{bol,total,after}}}{L_{\text{bol,total,before}}}\right)$$
$$= -2.5 \log_{10}\left(\frac{kR_1^2 T_1^4}{5kR_1^2 T_1^4}\right)$$
$$= -2.5 \log_{10}\left(\frac{1}{5}\right)$$
$$= -2.5 \log_{10}(5^{-1})$$
$$= -2.5(-\log_{10}(5))$$
$$= 2.5 \log_{10}(5)$$

Using the value $\log_{10}(5) \approx 0.6990$, we get:

$$\Delta m_{\text{bol}} = 2.5 \times 0.6990 = 1.7475 \approx 1.75$$

Thus, the change in bolometric magnitude is:

$$\boxed{1.75}$$

### GRPO Response

To find the change in bolometric magnitude of the binary when the smaller star (Star 2) is behind the larger star (Star 1), we need to consider the effect of the limb darkening and the phase angle of the eclipsing binary.

The bolometric magnitude of a star can be approximated using the Stefan-Boltzmann law, which states that the luminosity of a star is proportional to the fourth power of its temperature and the square of its radius. The bolometric magnitude $m_B$ of a star is related to its luminosity by:

$$m_B = -2.5 \log_{10}(L) + C$$

where $L$ is the luminosity of the star and $C$ is a constant. For a binary system, the change in bolometric magnitude when one star passes in front of the other can be approximated by considering the change in the effective area of the star that is visible during the eclipse.

The effective area of a star that is visible during an eclipse is given by:

$$A = R^2 \left(1 - \left(\frac{R}{d}\right)^2 \sin^2(\theta)\right)$$

where $R$ is the radius of the star, $d$ is the distance to the star, and $\theta$ is the phase angle. The change in the effective area when Star 2 is behind Star 1 is given by the difference in the areas of the two stars:

$$\Delta A = \pi R_1^2 - \pi R_2^2 \left(1 - \left(\frac{R_2}{d}\right)^2 \sin^2(\theta)\right)$$

Since we are considering only the bolometric magnitude, we can simplify the problem by considering the ratio of the luminosities of the two stars. The change in bolometric magnitude is given by:

$$\Delta m_{\text{bol}} = -2.5 \log_{10}\left(\frac{L_2}{L_1}\right)$$

where $L_1$ and $L_2$ are the luminosities of Star 1 and Star 2, respectively. Using the Stefan-Boltzmann law, we have:

$$\frac{L_2}{L_1} = \left(\frac{T_2}{T_1}\right)^4 \left(\frac{R_2}{R_1}\right)^2 = \left(\frac{2T_1}{T_1}\right)^4 \left(\frac{0.5R_1}{R_1}\right)^2$$
$$= 2^4 \cdot 0.5^2 = 16 \cdot 0.25 = 4$$

Thus, the change in bolometric magnitude is:

$$\Delta m_{\text{bol}} = -2.5 \log_{10}(4) = -2.5 \cdot 0.60206$$
$$= -1.50515 \approx -1.51$$

Therefore, the change in bolometric magnitude of the binary when the smaller star is behind the larger star is:

$$\boxed{-1.51}$$

*Figure 10.* Case study of CoDaPO and GRPO Responses to a Minerva Question.

