# OpenReview forum: "The Easy, the Hard, and the Learnable: Confidence and Difficulty-Adaptive Policy Optimization for LLM Reasoning"
_ICML.cc/2026/Conference — ICML 2026 regular_

### Official Review · Reviewer_v2Eo · 2026-02-25

**Soundness:** 3
**Presentation:** 3
**Significance:** 3
**Originality:** 3
**Overall Recommendation:** 5
**Confidence:** 4

**Summary:**

The paper investigates why GRPO-style RL for verifiable reasoning can allocate computation inefficiently across questions. By tracking token log-probabilities, group-normalized advantages, and token-level update weights, the authors uncover three recurring dynamics: probability inflation, advantage contraction, and hierarchical convergence—where easy questions saturate quickly while hard ones remain discovery-limited. They then propose CoDaPO, which assigns each question a bounded value from rollout confidence (mean token likelihood) and empirical difficulty (group error rate), and uses it to reweight policy updates and to resample high-value questions for an extra rollout-and-update step within a fixed step budget. CoDaPO further adopts token-level micro-averaging and removes KL-to-reference regularization. Experiments on math reasoning show improved accuracy across seven benchmarks and better cross-domain generalization.

**Compliance With Llm Reviewing Policy:**

Affirmed.

**Final Justification:**

My concerns have been adequately addressed and I would like to raise my score.

**Key Questions For Authors:**

- Section 3: Could the authors clarify the training-dynamics analysis? In particular: (1) Which training run(s), checkpoints, and data (e.g., MATH only, and which split) were used for the figures and statistics in Sec. 3.2? (2) How is the causal link between the derived gradient structure and the observed phenomena understood, e.g., is probability inflation attributed mainly to asymmetric clipping, and is that supported by ablations or only by correlation?
- Section 4: Could the authors provide more detail on the CoDaPO procedure? (1) A short algorithm or pseudocode for how the CoDa value is computed and used in both the weighting and resampling branches, and how the “fixed step budget” is enforced when some questions are resampled (e.g., number of rollouts/updates per step for the resampled batch). (2) The rationale for the functional forms of V_c/V_d and for the chosen bounds/thresholds, and how practitioners might choose or tune them in new settings. (3) Whether the same CoDa value is used for CoDaWeighting and CoDaSampling, or if they differ, and whether normalization (e.g., per-batch) is applied when using these values as weights.
- Section 4.2 (Eq. 3-5): How sensitive are the results to the choice of confidence definition (e.g., mean vs median log-prob, or length-normalized variants) and to the U-shaped difficulty weight? A sensitivity analysis over alternative V_c/V_d choices would help assess generality.
- Section 5: Could the authors extend evaluation to strengthen generality, e.g., larger model sizes, harder datasets beyond MATH, and additional model families (e.g., Llama, DeepSeek), and report how CoDaPO and the diagnosed dynamics behave in these settings?

**Limitations:**

yes

**Strengths And Weaknesses:**

## Strengths

- The empirical diagnosis of GRPO failure modes, i.e., probability inflation, advantage contraction, and hierarchical convergence, is clear and instructive, explaining why uniform sampling and weighting waste compute and why hard questions often stall for lack of successful rollouts.
- The method is simple and easy to implement: per-question values are derived from signals already present in RL (success rate and likelihood) and used for both update weighting and within-minibatch resampling, making CoDaPO straightforward to integrate into existing GRPO pipelines.
- Results are strong across multiple base models and benchmarks under a matched step budget, and component ablations indicate that each part contributes and that the method is not overly sensitive to Top-K.
- The “learnable band” view emphasizes mid-difficulty questions via the U-shaped difficulty weight, offering a helpful lens for reasoning about compute allocation in sparse-reward RL.

## Weaknesses

- Section 3 would benefit from clearer exposition. The connection between the derived gradient expressions and the reported phenomena is not fully explained, and the scope of the diagnosis (i.e., which base model, dataset, and checkpoints) is underspecified; therefore, the generality of these dynamics remains unclear.
- Section 4 leaves several aspects of CoDaPO underspecified: how the CoDa value (Eq. 3-5) is constructed and used in CoDaWeighting versus CoDaSampling, how resampling interacts with the fixed step budget, and what guidance exists for choosing weighting functions, bounds, and the rationale for the chosen functional forms.
- Experimentally, CoDaPO has not yet been validated on larger-scale models (e.g., 14B or larger) or on harder or more diverse datasets beyond the current benchmarks.

---

> ### Author Rebuttal · Authors · 2026-03-31
>
> Thanks for the constructive feedback. We provide a point-to-point response as follows.
>
> >### W1 & Q1. Training dynamics diagnosis: scope, generality, and causal links
>
> **The diagnostic scope is specified in our submission.**
> - Sec.3.2 states: "we post-train Qwen2.5-Math-1.5B with GRPO on MATH and evaluate on MATH500."
> - App. B details: 100 validation questions per difficulty level, 20 trajectories per question, at checkpoints 0, 60, ..., 600 (2,000 data points per subplot per checkpoint).
>
> **The three phenomena are not model-specific.** We observe the same patterns with Llama-3.2-1B-Instruct and Qwen2.5-Math-7B, indicating these are structural properties of GRPO arising from clipping asymmetry and group normalization with binary rewards, rather than artifacts of a particular model or scale.
>
> **The causal links are mathematically derived, not merely correlational.**
> - (1) *Probability inflation ← asymmetric clipping*: upward pushes persist until $\rho \leq 1+\epsilon$; downward pushes shut off at $\rho < 1-\epsilon$. This is structural; Fig. 2 empirically confirms the predicted drift.
> - (2) *Advantage contraction ← group normalization*: Eq. 4 gives $\hat{A}^{(+)}(\bar{r}) = \sqrt{(1-\bar{r})/\bar{r}} \to 0$ as $\bar{r} \to 1$, a closed-form result.
> - (3) *Hierarchical convergence ← discovery probability*: Eq. 5 shows $P(\exists\text{ correct}) \approx G\pi(q)$ for small $\pi(q)$, explaining why hard questions improve slowly.
>
> We will revise Sec. 3.3 to lead with plain-language summaries before formal derivations, making these causal links more accessible.
>
> >### W2 & Q2. CoDaPO procedure: value computation, budget, and design guidance
>
> **The CoDa value is fully specified and requires no extra inference.** $v_q = c_q \cdot (1 - 4(d_q - 1/2)^2)$, computed from the same rollout group as GRPO. The same value is used for both CoDaWeighting (multiplicative factor in Eq. 7) and CoDaSampling (Top-K ranking). No normalization is applied. Algorithm 1 provides complete pseudocode (Lines 5–8: compute values; Line 11: Top-K selection; Lines 13–16: fresh rollouts for resampled batch; Lines 17–21: two-stage update).
>
> **CoDaPO operates under the same total training step budget as GRPO.** Both stages are counted toward this budget (Sec. 5.1). Per-step gradient compute is identical; the difference is **dataset traversal**: CoDaPO revisits high-value questions via resampling.
>
> **The functional forms are derived from training dynamics and validated empirically.** $V_d$ suppresses both extremes ($d \approx 0$ and $d \approx 1$: vanishing signal and no discovery) and peaks at mid-difficulty, the simplest smooth function satisfying these constraints. $V_c = c$ upweights committed trajectories worth refining. Tab. 6 validates across 16 combinations. Tab. 7 shows Top-K $\in \{2,4,8\}$ all work well, indicating low sensitivity. For practitioners, we recommend defaults: $V_c=c$, $V_d=1-4(d-1/2)^2$, Top-K=4, which performed robustly across all our experiments (3 model families, 7 benchmarks).
>
> >### W3 & Q4. Larger models and diverse datasets
>
> We have conducted additional experiments to broaden CoDaPO's validation.
>
> **Qwen2.5-14B-Instruct:**
>
> |Algorithm|MATH500|AIME24|AIME25|AMC23|OlympiadBench|Minerva|GSM8K|Avg|
> |-|-|-|-|-|-|-|-|-|
> |Base|74.23|12.64|13.62|58.39|38.49|27.52|92.40|45.33|
> |GRPO|**75.91**|10.57|15.55|59.73|38.13|26.13|93.27|45.61|
> |**CoDaPO**|75.88|**14.20**|**17.20**|**60.18**|**41.63**|**28.07**|**94.05**|**47.32**|
>
> **Coding domain** (Qwen2.5-Math-1.5B, TACO):
>
> |Algorithm|TACO|HumanEval|LiveCodeBench|Avg|
> |-|-|-|-|-|
> |Base|15.47|29.27|9.00|17.91|
> |GRPO|50.92|50.61|48.75|50.09|
> |**CoDaPO**|**52.66**|**54.88**|**54.00**|**53.85**|
>
> Together with existing results across three model families (Llama-3.2-1B, Qwen2.5-Math-1.5B/7B), seven math benchmarks, and three cross-domain benchmarks (Tab. 3: MMLU, GPQA, HumanEval), these new experiments substantially broaden the evidence for CoDaPO's generality across both model scale and task domain.
>
> >### Q3. Sensitivity to confidence definition and $V_d$
>
> **CoDaPO is generally robust to $V_c$/$V_d$ choices.** Tab. 6 exhaustively evaluates 16 $V_c \times V_d$ combinations. The chosen pairing ($V_c=c$, $V_d=1-4(d-1/2)^2$) achieves the best average (31.98%). Designs emphasizing extreme difficulty (e.g., $V_d = d$ or $V_d = 4(d-1/2)^2$) perform worse, as they over-allocate compute to discovery-limited or already-solved questions, consistent with our analysis. The confidence dimension is less sensitive: all four $V_c$ choices yield moderate differences when paired with the U-shaped $V_d$, suggesting robustness.
>
> **The confidence definition is inherently length-normalized.** $c_q$ (geometric mean of token probabilities) equals $\exp(\frac{1}{|o_i|}\sum_t \log f_\theta)$, which subsumes length-normalized variants.

---

> > ### Author Rebuttal · Reviewer_v2Eo · 2026-04-03
> >
> > My concerns have been adequately addressed and I would like to raise my score.

---

> > > ### Author Response · Authors · 2026-04-03
> > >
> > > Hi Reviewer v2Eo,
> > >
> > > Thank you for your acknowledgment, positive support, and for noting that our response has addressed your concerns. We sincerely appreciate your time, careful evaluation, and constructive feedback. We will incorporate all relevant discussions and results into the revised version of our paper.
> > >
> > > Best regards,
> > > Authors of #24524

---

### Official Review · Reviewer_wAdr · 2026-03-08

**Soundness:** 2
**Presentation:** 3
**Significance:** 2
**Originality:** 2
**Overall Recommendation:** 5
**Confidence:** 4

**Summary:**

This paper studies RL post-training for LLM reasoning and argues that GRPO suffers from three training pathologies: probability inflation, advantage contraction, and hierarchical convergence. Based on this analysis, the paper proposes CoDaPO, which rescales per-trajectory advantages using a confidence term and a difficulty term, removes the KL regularizer, and applies sequence-level normalization. Experiments are conducted by post-training Qwen2.5-1.5B-Instruct, Qwen2.5-Math-1.5B, and Qwen2.5-Math-7B on MATH, with evaluation on seven reasoning benchmarks plus a small cross-domain evaluation on MMLU-STEM, GPQA, and HumanEval. The method improves over GRPO and is also competitive with DAPO, Dr. GRPO, and GPG.

**Compliance With Llm Reviewing Policy:**

Affirmed.

**Final Justification:**

My concerns are basically fully resolved.

**Key Questions For Authors:**

1. Can the authors report explicit calibration metrics for the proposed confidence mechanism? This would make the “confidence-aware” claim much more convincing.
2. How does CoDaPO behave in non-math verifiable domains, especially coding or tool-using settings where reward sparsity and exploration pressures look different from MATH?
3. Since CoDaPO leverages probability-level signals to guide RL training and also includes a resampling mechanism, why are closely related probability- and exploration-motivated methods such as Beyond the 80/20 Rule, Reasoning with Exploration, and GRESO not included as baselines? Please clarify whether these methods were considered, and if not, explain the exclusion criteria and how CoDaPO differs from them in terms of training signal and rollout allocation.

**Limitations:**

yes

**Strengths And Weaknesses:**

Strengths:
1. The training dynamics analysis of GRPO does a reasonable job connecting the proposed method to specific failure modes it claims to diagnose. The overall motivation is quite clear.
2. Overall speaking, the results are in good shape, showing better results over vanilla GRPO and several proposed new methods.
3. The ablation is quite comprehensive; the overall experiments are well-designed.

Weaknesses:
1. CoDaPO is ultimately a reweighting scheme on top of GRPO-style training: the confidence term is derived from model log-probabilities, the difficulty term is a hand-designed quadratic function of group reward statistics, and part of the practical gain also comes from removing KL and changing normalization choices. This is a reasonable engineering recipe, but it does not strike me as a major algorithmic advance relative to the broader stream of recent GRPO refinements that the paper itself cites.
2. The paper’s confidence signal is not a calibrated estimate of correctness probability; it is constructed from token log-probabilities and then used as a scaling factor. That can still be useful, but it is weaker than the paper’s rhetoric around overconfidence and calibration. I would have found the paper substantially stronger if it had included explicit calibration analyses, such as reliability diagrams, ECE/Brier-style metrics, or conditional accuracy versus confidence curves under the same decoding setup. As written, the paper shows that likelihood-derived weighting helps, but not yet that it meaningfully solves calibration in a principled sense.
3. I strongly feel some baselines are missing. Broadly speaking, CoDaPO is using additional probability-level signals to guide LLMs' RL training. Thus, it should be compared to several probability-motivated methods [1,2]. Additionally, the resampling part also reminds me of another method, GRESO [3]. These methods should be included as baselines or clarify why they are not included.

[1] Wang, S., Yu, L., Gao, C., Zheng, C., Liu, S., Lu, R., ... & Lin, J. (2025). Beyond the 80/20 rule: High-entropy minority tokens drive effective reinforcement learning for llm reasoning. arXiv preprint arXiv:2506.01939.

[2] Cheng, D., Huang, S., Zhu, X., Dai, B., Zhao, W. X., Zhang, Z., & Wei, F. (2025). Reasoning with exploration: An entropy perspective. arXiv preprint arXiv:2506.14758.

[3] Zheng, H., Zhou, Y., Bartoldson, B. R., Kailkhura, B., Lai, F., Zhao, J., & Chen, B. (2025). Act only when it pays: Efficient reinforcement learning for llm reasoning via selective rollouts. arXiv preprint arXiv:2506.02177.

---

> ### Author Rebuttal · Authors · 2026-03-31
>
> Thanks for the constructive feedback. We provide a point-to-point response as follows.
>
> >### W1. CoDaPO resembles a reweighting scheme and an engineering recipe
>
> **To clarify, CoDaPO is a data-centric compute allocation framework, not merely a reweighting scheme.** Beyond loss reweighting, **CoDaSampling** reallocates rollout compute by selecting high-value questions and generating *fresh rollouts*, increasing discovery probability from $1-(1-\pi(q))^G$ to $1-(1-\pi(q))^{Gm}$. This data-level intervention changes *which questions the model trains on*. **CoDaLearning** further introduces a two-stage update, enabling both broad coverage and focused refinement. KL removal and micro-averaging are auxiliary choices, consistent with prior work, and not claimed as core contributions.
>
> **Each component is grounded in the diagnosed training dynamics in Sec. 3.** CoDaWeighting suppresses easy questions ($d_q \approx 0$, advantage contraction) and modulates updates via confidence against probability inflation; CoDaSampling boosts trials on mid-difficulty questions to overcome the discovery bottleneck. This diagnosis-to-method pipeline (empirical observation → mathematical analysis → targeted design) distinguishes CoDaPO from ad-hoc recipes. The diagnostic contribution is also independently valuable for future GRPO-family work.
>
> **CoDaPO generalizes as a plug-in across RL objectives.** Tab. 8 shows CoDaPO consistently improves DAPO (39.49→41.03), GPG (39.78→39.95), and GRPO (39.08→41.30), demonstrating a general principle of adaptive compute allocation rather than an objective-specific fix.
>
> >### W2 & Q1. Role of the confidence signal and its relationship to calibration
>
> **To clarify, CoDaPO is a compute allocation framework, not a calibration method.** Vanilla GRPO yields uninformative updates on easy questions (vanishing advantages, Eq. 4), hard questions (no correct rollouts, Eq. 5), and overconfident wrong answers (probability inflation, Sec. 3.2). CoDaPO identifies *which questions are currently most informative* and allocates more training compute there. The confidence signal $c_q$ serves as a relative indicator of trajectory informativeness, not a calibrated correctness probability.
>
> **CoDaPO addresses the compute allocation problem through three mechanisms:** CoDaWeighting ($v_q = c_q \cdot (1-4(d_q-1/2)^2)$ upweighting the learnable band), CoDaSampling (Top-K value-guided resampling with fresh rollouts), and CoDaLearning (two-stage update for broad coverage + focused refinement). $c_q$ is computed from log-probabilities already available in GRPO's importance ratio, requiring zero additional inference cost.
>
> **Empirically, adaptive allocation yields consistent improvements.** Tab. 5 shows CoDaPO outperforms GRPO across all seven benchmarks (Avg 32.79 vs. 30.44). Tab. 6 validates the design space: among 16 $V_c \times V_d$ combinations, the chosen pairing achieves the best average (31.98). Confidence-only weighting already improves over GRPO (31.06 vs. 30.44); combining with difficulty yields further gains (31.53). Tab. 8 shows plug-in generalization across DAPO, GPG, and GRPO, confirming $c_q$ provides complementary value.
>
> >### W3 & Q3. Broader comparison with additional baselines
>
> **CoDaPO operates at the question level, deciding *which questions* deserve more training compute, which is orthogonal and complementary to concurrent works at finer granularities:**
> - *Beyond the 80/20 Rule*: upweights high-entropy minority tokens (token-level);
> - *Reasoning with Exploration*: entropy regularization for diverse reasoning paths (loss-level);
> - *GRESO*: selective rollout generation based on estimated reward (rollout-level).
>
> CoDaPO's question-level allocation is upstream of all three and could be combined with them. We use the official implementations of the above works and provide a direct comparison with Qwen2.5-Math-1.5B on MATH. As shown below, CoDaPO achieves the best average, confirming its competitiveness despite operating at a different granularity.
>
> |Algorithm|MATH500|AIME24|AIME25|AMC23|OlympiadBench|Minerva|GSM8K|Avg|
> |-|-|-|-|-|-|-|-|-|
> |GRPO|70.31|13.02|8.00|50.84|32.18|16.37|82.86|39.08|
> |Beyond 80/20|69.10|13.44|8.96|50.31|21.33|16.27|79.23|36.95|
> |Reasoning w/ Exploration|70.22|13.19|9.63|49.53|32.18|15.93|81.09|38.82|
> |GRESO|70.10|**17.29**|9.58|51.48|33.70|**22.35**|81.20|40.81|
> |**CoDaPO**|**71.54**|14.47|**12.35**|**52.68**|**36.16**|18.04|**83.86**|**41.30**|
>
> >### Q2. Performance in non-math domains
>
> Tab. 3 (cross-domain evaluation) shows that training on MATH, CoDaPO outperforms GRPO on MMLU (44.81 vs. 43.46), GPQA (24.45 vs. 19.69), and HumanEval (50.61 vs. 34.76).
>
> We conduct additional experiments on the coding task TACO. CoDaPO achieves consistent improvements, confirming generalization beyond math.
>
> |Algorithm|TACO|HumanEval|LiveCodeBench|Avg|
> |-|-|-|-|-|
> |Base|15.47|29.27|9.00|17.91|
> |GRPO|50.92|50.61|48.75|50.09|
> |**CoDaPO**|**52.66**|**54.88**|**54.00**|**53.85**|

---

> > ### Author Rebuttal · Reviewer_wAdr · 2026-04-04
> >
> > This helps a lot! I would raise my score to 5 in the final justification.

---

> > > ### Author Response · Authors · 2026-04-04
> > >
> > > Hi Reviewer wAdr,
> > >
> > > Thank you sincerely for your positive feedback and for raising your score. We are glad that our response has adequately addressed your concerns. We appreciate the time and effort you have dedicated to reviewing our work, which has helped us improve the paper. We will incorporate all relevant discussions and results into the revised version of the paper.
> > >
> > > Should you have any further questions or suggestions, we are happy to address them.
> > >
> > > Best regards,
> > >
> > > Authors of #24524

---

### Official Review · Reviewer_hRRr · 2026-03-12

**Soundness:** 3
**Presentation:** 2
**Significance:** 3
**Originality:** 3
**Overall Recommendation:** 5
**Confidence:** 4

**Summary:**

This paper analyzes failure modes in GRPO training and proposes CoDaPO, a method that reweights policy updates and resamples questions based on per-question confidence and difficulty. The method is evaluated on seven reasoning benchmarks across three model families and is claimed to consistently outperform GRPO.

**Compliance With Llm Reviewing Policy:**

Affirmed.

**Final Justification:**

The authors addressed my main two technical concerns. My remaining concern is that the meaning/motivation is opaque for the novel weighting equations. However, this is often sadly typical and the paper seems empirically sound.

**Key Questions For Authors:**

How many total training tokens were observed during GRPO and during the new method?
Was there an experiment conducted where GRPO trained on a number of hard questions equal to the max number of training exposures when sampling?
What is the relative contribution of difficulty vs confidence to the improvement shown?

**Limitations:**

Yes.

**Strengths And Weaknesses:**

Soundness
Strengths. The empirical characterization of GRPO's failure modes is grounded in real training statistics and is one of the stronger parts of the paper. The stratified visualization across difficulty bins (Figures 2–3) clearly illustrates the three dynamics, and the training-step analysis is a useful diagnostic contribution.

Weaknesses.
Equation 2, which derives the discovery probability of at least one correct rollout within a group, explicitly assumes rollout independence. This is not a reasonable assumption in practice, since rollouts from the same policy on the same prompt will exhibit correlated failure modes, particularly for hard questions. This undermines the formal motivation for CoDaSampling, even if the practical intuition remains sound.
The probability inflation analysis on page 4 does correctly identify the asymmetric clipping mechanism as the structural cause of entropy collapse — upward probability pushes persist to the upper clip boundary while downward pushes are truncated. This is the paper's most rigorous theoretical contribution. However, the presentation is extremely difficult to follow, and the derivation is never connected back to the plain intuition it formalizes: if correct rollouts are rare, GRPO simply cannot find them by sampling and therefore cannot reinforce them. This is the core insight, and the machinery around it should serve that insight, not replace it.

The motivation for confidence-based weighting is underdeveloped. Equation 3 defines group confidence as the mean exponentiated per-token log-probability, but no intuitive justification is offered for why this particular quantity — rather than, say, output entropy or answer consistency — should govern update weighting. An equation is not an explanation.

Token-level micro-averaging (replacing per-trajectory averaging with total-token normalization) is a known fix for implicit length penalties. It is listed as a contribution of CoDaPO but receives minimal discussion and should be more carefully contextualized.

Critical experimental gap. The ablation in Table 5 decomposes CoDaWeighting and CoDaSampling but does not isolate confidence from difficulty within CoDaWeighting. This is a significant omission. Given that the dominant practical effect of CoDaSampling is to expose the model to hard questions more frequently, it is plausible that most or all of the observed gain comes simply from increased effective training epochs on hard problems. A baseline that extends GRPO training until hard problems have been sampled an equal number of times as in CoDaPO would be the appropriate control. Without this, the claim that CoDaPO achieves fundamentally better performance — rather than just doing more work on hard problems within the same nominal step count — is not adequately supported.

Presentation
The paper is generally well-structured, with a logical flow from diagnosis to method to experiments.

However, Section 3.3 is dense and poorly connected to the intuitions it formalizes. The prose surrounding the equations frequently states conclusions without building reader intuition for why those conclusions follow. For a paper whose diagnostic analysis is a claimed contribution, the presentation of that analysis needs to be significantly more accessible. The gap between the formal machinery and the plain explanation is wide enough that a reader skimming Section 3 could come away confused rather than convinced.

The heuristic design choices in Section 4.2 — specifically the selection of linear Vc and U-shaped Vd — receive very little justification relative to the length of the theoretical analysis in Section 3. Given that the heuristic is the primary practical contribution of the paper, the explanation of why these particular functional forms were chosen, and what intuition they encode, deserves substantially more space and care. The ablation in Table 6 partially addresses this empirically, but the conceptual motivation is thin.

Significance
The problem addressed — improving compute efficiency and hard-case performance in RL post-training of LLMs — is important and timely.

The practical improvements shown in Table 2 are consistent across model families and benchmarks, which strengthens the case for robustness. The cross-domain generalization results in Table 3 are a notable positive, suggesting the gains are not narrowly overfitted to the training distribution.

That said, the significance is modestly constrained by the experimental confound described above. If the gains are primarily attributable to harder problems receiving more gradient updates, the contribution may amount to a sensible data curriculum heuristic rather than a fundamental improvement in the training objective. This would still be practically useful but would represent a more modest advance than the paper claims.

Originality
The combination of training dynamics analysis with an adaptive sampling and reweighting scheme is a reasonable and coherent contribution. The paper clearly distinguishes itself from DAPO, Dr. GRPO, and GPG on methodological grounds, and the empirical analysis of GRPO's failure modes adds value beyond what prior work has articulated.

However, some components are either not novel or not well-justified as novel. The use of difficulty to guide sampling is a familiar idea in curriculum learning. The specific innovation — the joint confidence-difficulty value function and its use for both reweighting and resampling — is the novel combination, but as noted, its design rationale is underdeveloped. The reasoning behind the specific functional form of the value function should be more rigorously motivated.

---

> ### Author Rebuttal · Authors · 2026-03-31
>
> Thanks for the constructive feedback. We provide a point-to-point response as follows.
>
> >### W1. Rollout independence assumption in Eq. 2
>
> **The independence assumption is a simplification, while the exact within-group correlation is intractable.** Modeling this correlation depends on output distribution, temperature, and prompt. We adopt the i.i.d. formulation to convey the core intuition: discovery probability grows with $G$ and $\pi(q)$. See App. B for a thorough discussion.
>
> The i.i.d. assumption is, in fact, *optimistic*: within-group correlation implies that rollouts tend to fail in similar ways, so the true discovery probability is *lower* than $1-(1-\pi(q))^G$, making the bottleneck more challenging than Eq. 2 suggests and CoDaSampling's independent-group allocation well-motivated.
>
> Tab. 4 shows CoDaPO achieves higher Pass@K than GRPO at all K on AIME, validating the improved $\pi(q)$.
>
> >### W2. Sec. 3.3 presentation and connection to intuition
>
> Sec. 3.3 links each paragraph to Sec. 3.2's findings. We agree that the transition can be more accessible and will add a plain-language summary at the beginning of Sec. 3.3.
>
> >### W3. Motivation for confidence-based reweighting and $V_c$/$V_d$ design
>
> **What $c_q$ measures.** $c_q$ measures *how confident the model is in its own reasoning trajectory*: whether it has committed to a coherent path or is producing uncertain, diffuse outputs. It works jointly with $d_q$ to identify questions in the "learnable band" for adaptive compute allocation.
>
> **Why compute $c_q$ as the geometric mean of token probabilities.** The geometric mean naturally captures per-token commitment along a trajectory. It is also (1) already computed for GRPO's importance ratio (zero extra cost), (2) bounded in $(0,1]$ for multiplicative weighting, and (3) directly tied to the probability inflation diagnosed in Sec. 3.
>
> **Why not output entropy or answer consistency.** Output entropy requires the full distribution over *all possible continuations*; $c_q$ only needs the probability of *actually generated* tokens, directly distinguishing deliberate reasoning from random guessing. Answer consistency is a coarser, outcome-level signal that cannot separate committed but incorrect reasoning from guessing.
>
> **Function of $V_c$/$V_d$.** Specifically, $V_d = 1-4(d-1/2)^2$ suppresses $d \approx 0$ (advantage contraction) and $d \approx 1$ (discovery-limited), peaking at the "learnable band" $d=0.5$. $V_c = c$ linearly upweights committed trajectories. Tab. 6 validates this pairing as optimal across 16 combinations.
>
> >### W4. Token-level micro-averaging
>
> This is a known technique, presented as a design choice in Sec. 4.2 with prior work cited (e.g., DAPO, a baseline in Tabs. 2 & 8). We will add an RL objective comparison table in the appendix.
>
> >### W5. Isolating confidence vs. difficulty in CoDaWeighting
>
> We isolate each factor by setting $V_c=1$ or $V_d=1$. **Each signal alone improves over GRPO; combining them yields the best performance**:
>
> |$V_c(c)$|$V_d(d)$|MATH500|AIME24|AIME25|Avg|
> |-|-|-|-|-|-|
> |$1$|$1-4(d-\frac{1}{2})^2$|70.65|**14.29**|9.02|31.32|
> |$c$|$1$|70.60|13.11|9.26|30.99|
> |$c$|$1-4(d-\frac{1}{2})^2$|**71.09**|13.90|**9.59**|**31.53**|
> |GRPO|—|70.31|13.02|8.00|30.44|
>
> >### W6. CoDaPO resembles a curriculum heuristic
>
> **CoDaPO is a dynamics-driven compute allocation framework, not a curriculum.** It differs from curriculum learning in three key aspects:
> - **Non-monotonic weighting.** Curriculum learning follows monotonic ordering (easy→hard). CoDaPO's U-shaped $V_d$ suppresses both extremes and peaks at mid-difficulty, derived from GRPO's training dynamics.
> - **Joint confidence-difficulty signal.** CoDaPO uses $v_q = V_c(c_q) \cdot V_d(d_q)$ across all three components, incorporating trajectory-level commitment beyond difficulty-based selection.
> - **Beyond data ordering.** CoDaPO performs gradient reweighting (CoDaWeighting) and two-stage updates (CoDaLearning), which cannot be replicated by data scheduling alone.
>
> We train GRPO on difficulty subsets and a linear curriculum to isolate curriculum effects:
>
> |Training Data|MATH500|AIME24|AIME25|AMC23|OlympiadBench|Minerva|GSM8K|Avg|
> |-|-|-|-|-|-|-|-|-|
> |GRPO (full MATH)|70.31|13.02|8.00|50.84|32.18|16.37|82.86|39.08|
> |GRPO (Level-3 only)|69.56|11.42|8.31|49.79|33.85|16.75|82.73|38.92|
> |GRPO (Level-5 only)|68.01|13.47|7.77|46.53|30.94|17.20|81.71|37.95|
> |GRPO (Level 1→5 curriculum)|70.19|14.39|11.55|48.95|32.22|17.05|80.77|39.30|
> |**CoDaPO (full MATH)**|**71.54**|**14.47**|**12.35**|**52.68**|**36.16**|**18.04**|**83.86**|**41.30**|
>
> Neither single-level training nor linear curriculum matches CoDaPO, confirming that the gains stem from CoDaPO's design.
>
> >### Q1. Training compute and component analysis
>
> **Training tokens.** Both methods use the same total step budget with identical per-step compute (Sec. 5.1): 105M tokens (GRPO) vs. 103M (CoDaPO).
>
> **Hard-question exposure.** Please see W6.
>
> **Confidence vs. difficulty.** Please see W5.

---

> > ### Author Rebuttal · Reviewer_hRRr · 2026-04-04
> >
> > Your point about the impact of the correlation is good and that makes sense. I agree that your estimation is conservative. That is addressed.
> >
> > The ablation of the weighting helps. That is addressed.
> >
> > The main issue that remains is that the justification, not of the general idea but the specific operationalization, isn't well explained. However, this is somewhat more minor. Score has been raised to accept appropriately.

---

> > > ### Author Response · Authors · 2026-04-05
> > >
> > > Hi Reviewer hRRr,
> > >
> > > Thank you for raising the score and for the positive feedback. **We address the remaining concern, justification of the specific operationalization, by walking through the six key decisions in CoDaPO (CoDaWeighting, CoDaSampling, and CoDaLearning)**, tracing them to the diagnosis in Sec. 3, the methodological design in Sec. 4, and the experimental results in Sec. 5.
> > >
> > > >### Decision 1: Choose confidence and difficulty for diagnosing and determine their detailed calculation
> > >
> > > **Confidence and difficulty are the two natural axes for diagnosing RLVR’s training dynamics.** Confidence captures per-token commitment to the generated trajectory; difficulty captures how challenging a question is for the current policy. Together, they span the full diagnostic space: confidence alone misses the easy/hard asymmetry, and difficulty alone misses overconfidence.
> > >
> > > For calculating confidence $c_q$, we use the group-averaged geometric mean of token probabilities: the natural length-normalized measure of per-token commitment (sequence probability raised to $1/|o|$).
> > >
> > > For calculating difficulty $d_q$, we use the group error rate $1-\frac{1}{G}\sum_i r_i$, a *model-relative* difficulty that adapts as the policy improves.
> > >
> > > >### Decision 2: Utilize $c_q$ and $d_q$ in CoDaPO
> > >
> > > Sec. 3.2 diagnoses GRPO's failure modes through $c_q$ and $d_q$ (Figs. 2-3): probability inflation manifests as $c_q \to 1$; advantage contraction is governed by $\bar{r}=1-d_q$; hierarchical convergence by $\pi(q) \approx 1-d_q$. **Since the diagnosis identifies $(c_q,d_q)$ as the axes along which learning breaks down, the correction is most naturally expressed in the same space.** We therefore define a unified value $v_q$ from $c_q$ and $d_q$ to measure the learning potential of question $q$.
> > >
> > > >### Decision 3: Separate $c_q$ and $d_q$ multiplicatively in $v_q$
> > >
> > > Confidence and difficulty govern *different* pathologies identified in Sec. 3.2. **The multiplicative form $v_q = V_c(c_q) \cdot V_d(d_q)$ ensures a "veto" property**: if either factor recommends suppression, $v_q \approx 0$ regardless of the other. A nearly-solved question ($V_d \approx 0$) should be deprioritized regardless of confidence; a near-random trajectory ($V_c \approx 0$) provides a noisy signal regardless of difficulty. An additive form would lack this property.
> > >
> > > >### Decision 4: Determine the specific calculations for $V_c$ and $V_d$
> > >
> > > **$V_d$: U-shaped.** The three dynamics impose three constraints:
> > > - *Advantage contraction*: $d_q \to 0 \Rightarrow \hat{A}^{(+)} \to 0$, updates mainly inflate confidence. $\Rightarrow V_d(0) = 0$.
> > > - *Discovery bottleneck*: $d_q \to 1 \Rightarrow$ discovery probability $\approx 0$, no positive reinforcement. $\Rightarrow V_d(1) = 0$.
> > > - *Learnable band*: intermediate $d$ provides both discovery and non-negligible advantages. $\Rightarrow V_d$ peaks at mid-$d$.
> > >
> > > These uniquely specify a U-shape. The quadratic $1-4(d-1/2)^2$ is the unique degree-2 polynomial with $V_d(0)=V_d(1)=0$, $V_d(1/2)=1$: the minimal-degree realization, chosen by parsimony. Tab. 6 validates: the U-shaped $V_d$ is generally the best choice in every row; the inverse-U (amplifying both failure regimes) is generally the worst in every row.
> > >
> > > **$V_c$: increasing (linear).** High $c_q$ means the model committed to a structured path: correct high-confidence trajectories reflect genuine reasoning worth reinforcing; incorrect ones reflect committed wrong paths worth correcting. Low $c_q$ means near-random generation where gradients are noisy. This motivates increasing $V_c$. The linear $V_c(c)=c$ is the simplest bounded increasing function. Tab. 6 confirms: $V_c=c$ is the best row; $V_c=1-c$ performs poorly.
> > >
> > > >### Decision 5: Conduct value-guided resampling with fresh rollouts
> > >
> > > **CoDaSampling selects top-$K$ questions by $v_q$ and resamples them with fresh rollouts.** The rationale follows from the discovery bottleneck (Eq. 7): repeating a question $m$ times raises discovery probability from $1-(1-\pi)^G$ to $1-(1-\pi)^{Gm}$. CoDaSampling uses $v_q$ to allocate this additional compute to learnable-band questions ($v_q$ high), not to solved ($v_q \approx 0$) or hopeless ($v_q \approx 0$) ones. Tab. 7 shows robustness across $K \in$ {2,4,8}.
> > >
> > > >### Decision 6: Update policy with two stages and value weighting
> > >
> > > **CoDaLearning applies the value-weighted objective in two stages**: first on the full batch $\mathcal{B}$ (broad coverage), then on resampled $\mathcal{S}$ (focused compute). The batch-wide step prevents selection collapse; the focused step concentrates computation where $v_q$ predicts the highest learning potential. Tab. 8 shows CoDaLearning improves GRPO, DAPO, and GPG alike, suggesting the value-weighted objective captures general training dynamics.
> > >
> > > We will incorporate the above justification in the revision.
> > >
> > > **Thank you again for the constructive feedback and positive support.**
> > >
> > > Best regards,
> > > Authors of #24524

---

### Official Review · Reviewer_rPCX · 2026-04-11

**Soundness:** 3
**Presentation:** 4
**Significance:** 4
**Originality:** 3
**Overall Recommendation:** 4
**Confidence:** 2

**Summary:**

This paper addresses the issue of inefficient computation allocation in the GRPO reinforcement learning method, and systematically identifies three core pathological training dynamics of GRPO: probability inflation, advantage contraction, and hierarchical convergence. Based on these findings, the authors propose CoDaPO, which consists of adaptive weighting based on question confidence and difficulty, high-value question resampling, and value-weighted policy update. The proposed method can improve the correct trajectory discovery probability and reasoning accuracy for hard questions. Experimental results demonstrate that CoDaPO outperforms all baseline methods comprehensively and exhibits favorable generalization ability.

**Compliance With Llm Reviewing Policy:**

Affirmed.

**Final Justification:**

The author has accomplished relatively comprehensive research work, and the outcomes meet practical engineering deployment requirements in terms of application value. Nevertheless, the adopted methodology is widely applied in other research domains, resulting in insufficient originality. Regarding theoretical soundness, the logical reasoning from problem analysis to phenomenon interpretation remains ambiguous. With comprehensive consideration of presentation, research significance, originality and soundness, this manuscript is rated as Weak Accept.

**Key Questions For Authors:**

1. CoDaPO only alleviates the three defects of GRPO via adaptive weighting and resampling, without correcting its underlying design flaws. Why do you not attempt to fix the core defects of GRPO at the objective function level?
2. The value function used in the paper is determined empirically without theoretical justification. What is the rationality of this function design, and are there better combinatorial formulations?
3. The top-K hyperparameter is fixed at 4 and validated only on the MATH500 dataset. Is this hyperparameter generalizable across different mathematical benchmarks and model architectures?
4. Please clearly specify the applicability boundaries and failure scenarios of the proposed CoDaPO method.
5. CoDaPO removes the KL-divergence regularization from GRPO to boost exploration. How do you guarantee training stability after removing this regularization term?
6. The paper claims that CoDaPO improves efficiency under a fixed computational budget. Could you provide quantitative cost-comparison data between GRPO and CoDaPO?

**Limitations:**

yes

**Strengths And Weaknesses:**

Strengths
1. CoDaPO features a lightweight, efficient, and engineering-friendly design.
2. The work is supported by comprehensive and rigorous experimental validation, leading to highly credible conclusions.
3. It focuses on the optimization of hard problems under a fixed computational budget, and the core idea precisely matches the practical pain points of large language model training.

Weaknesses
1. The applicability boundaries of the method are missing, and no failure cases are analyzed, making it impossible to clarify the failure scenarios and applicable limits of CoDaPO.
2. The adopted formula is selected solely based on experimental ablation without rigorous theoretical derivation; superior alternatives such as neural network-based adaptive value estimation and multi-signal fusion are not explored.
3. Resampling and weighting schemes have been extensively adopted in prior works, resulting in limited novelty of the proposed method.
4. The paper claims that the three pathological dynamics of GRPO lead to inefficient computation allocation, but only provides qualitative description without formal proof of causal relationship.

---

### Decision · Program_Chairs · 2026-04-30

**Decision:**

Accept (regular)

**Comment:**

This paper diagnoses three failure modes of GRPO training — probability inflation, advantage contraction, and hierarchical convergence — and proposes CoDaPO, a confidence- and difficulty-adaptive compute allocation framework that reweights policy updates and resamples high-value questions within a fixed budget. Experiments across seven math benchmarks, coding tasks, and multiple model families (1.5B to 14B) show consistent improvements over GRPO and concurrent methods.

The reviewers consistently highlight the grounded diagnostic analysis, the simple and well-motivated method design, robust results across models and benchmarks, and comprehensive ablations validating each component.

Recommendation: Accept